# Neuronal activity patterns in microcircuits of the cerebellar cortical C3 zone during reaching

Nadia L. Cerminara[1] , Martin Garwicz[2], Henry Darch[1], Conor Houghton[3] , Dilwyn E. Marple-Horvat[4] and Richard Apps[1]

[1] *School of Physiology, Pharmacology and Neuroscience, University of Bristol, Bristol, UK*
[2] *Neuronano Research Centre and Birgit Rausing Centre for Medical Humanities, Department of Experimental Medical Science, Lund University, Lund, Sweden*
[3] *Department of Computer Science, University of Bristol, Bristol, UK*
[4] *Department of Life Sciences, Manchester Metropolitan University, Manchester, UK*

Handling Editors: Richard Carson & Vatsala Thirumalai

The peer review history is available in the Supporting Information section of this article (https://doi.org/10.1113/JP282928#support-information-section).

**Nadia Cerminara** is a Research Fellow at the University of Bristol, UK. She obtained her PhD in neurophysiology from Monash University, Australia, in 2002. Using an *in vivo* systems-level approach, a key aim of her work is to determine the information processing that occurs in cerebellar circuits during the performance of both new and well-rehearsed movements.

N. L. Cerminara, M. Garwicz, D. E. Marple-Horvat and R. Apps contributed as senior authors.
This article was first published as a preprint. Cerminara NL, Garwicz M, Darch H, Houghton C, Marple-Horvat DE, Apps R. 2020. Action-based organization and function of cerebellar cortical microcircuits. bioRxiv. https://doi.org/10.1101/2020.04.04.025387

**Abstract** The cerebellum is the largest sensorimotor structure in the brain. A fundamental organizational feature of its cortex is its division into a series of rostrocaudally elongated zones. These are defined by their inputs from specific parts of the inferior olive and Purkinje cell output to specific cerebellar and vestibular nuclei. However, little is known about how patterns of neuronal activity in zones, and their microcircuit subdivisions, microzones, are related to behaviour in awake animals. In the present study, we investigated the organization of microzones within the C3 zone and their activity during a skilled forelimb reaching task in cats. Neurons in different microzones of the C3 zone, functionally determined by receptive field characteristics, differed in their patterns of activity during movement. Groups of Purkinje cells belonging to different receptive field classes, and therefore belonging to different microzones, were found to collectively encode different aspects of the reach controlled by the C3 zone. Our results support the hypothesis that the cerebellar C3 zone is organized and operates within a microzonal frame of reference, with a specific relationship between the sensory input to each microzone and its motor output.

(Received 3 February 2022; accepted after revision 7 October 2022; first published online 26 October 2022)

**Corresponding author** Nadia L. Cerminara: School of Physiology, Pharmacology and Neuroscience, University of Briso, University Walk, Bristol BS8 2SE, UK.    Email: n.cerminara@bristol.ac.uk

**Abstract figure legend** A defining feature of cerebellar organization is its division into a series of zones and smaller subunits termed microzones. The latter are thought to represent the basic operational unit of the cerebellum in its control of behaviour. Much of how zones and microzones are organized has been determined in anaesthetized preparations, and little is known about their function in awake animals. To address this, we recorded from neurons in the forelimb part of the C3 zone by recording from single cerebellar cortical neurons located in different microzones during a forelimb reach–retrieval task in cats. Neurons from individual microzones had characteristic patterns of activity during movement that collectively are consistent with operating within a movement frame of reference.

## Key points

- A defining feature of cerebellar organization is its division into a series of zones and smaller subunits termed microzones.
- Much of how zones and microzones are organized has been determined in anaesthetized preparations, and little is known about their function in awake animals.
- We recorded from neurons in the forelimb part of the C3 zone 'in action' by recording from single cerebellar cortical neurons located in different microzones defined by their peripheral receptive field properties during a forelimb reach–retrieval task in cats.
- Neurons from individual microzones had characteristic patterns of activity during movement, indicating that function is organized in relation to microcomplexes.

## Introduction

The cerebellum is widely recognized as a 'sensorimotor' structure, but how that translates to functional organization and neuronal activity at a microcircuit level during behaviour remains unclear. Fundamental to its mode of operation is its structural and functional organization into a series of 'modules'. Anatomically, each module is defined by its climbing fibre afferents from a specific subdivision of the inferior olive, targeting one or more rostrocaudally oriented cerebellar cortical zones (e.g. Atkins & Apps, 1997; Buisseret-Delmas & Angaut, 1993; Cerminara et al., 2013; Garwicz et al., 1996; Pijpers et al., 2006; Voogd & Bigare, 1980), and by output from Purkinje cells located within each zone, providing a highly convergent efferent projection to specific territories within the cerebellar and vestibular nuclei (Apps & Garwicz, 2000; Cerminara et al., 2013; Dietrichs, 1983; Garwicz et al., 1996; Sugihara et al., 2009; Voogd & Ruigrok, 2004).

High-resolution electrophysiological mapping has revealed that the zonal component of at least some individual modules is divided into smaller units termed 'microzones'. These are rostrocaudally extended groups of Purkinje cells that have similar climbing fibre peripheral receptive fields (Andersson & Eriksson, 1981; Andersson & Oscarsson, 1978; Oscarsson, 1973; Oscarsson & Uddenberg, 1966). Microzones and their associated olivo-cortico-nuclear microcircuits (collectively termed microcomplexes; Ito, 1984) are widely considered the fundamental operational units of the cerebellum

(Andersson & Oscarsson, 1978; Apps & Garwicz, 2005; Ito, 1984; Oscarsson, 1973; Oscarsson, 1979). However, evidence to support or challenge this concept in the awake behaving animal is surprisingly limited.

Microzones have been studied most extensively in the paravermal C3 zone, in the anaesthetized cat (Ekerot et al., 1991a; Jorntell et al., 1996). Quantitative analyses of the spatial organization of receptive fields using natural stimulation of the skin, and extended to include patterns of convergence between cutaneous and muscle afferents, identified 10 classes of receptive fields that could be further divided into over 40 subclasses (microzones) within the forelimb area of the zone. These analyses strongly suggested that the information conveyed by climbing fibre afferents is not purely 'sensory' but relates to the activity in spinal reflex circuits acting on single forelimb muscles (Garwicz 2002; Garwicz et al, 2002). Moreover, the spatial distribution of cutaneous receptive fields of climbing fibres is consistent with signalling how the skin surface moves in space upon contraction of single forelimb muscles. Since mossy fibres, the other main type of afferents to the cerebellar cortex, have similar receptive fields to climbing fibres in the C3 zone, it has been proposed that detection of the effects of elemental movements is a general feature of the input to this zone.

Studies in decerebrate ferrets of Purkinje cell activity during conditioning of reflex responses in the C3 'eyeblink' microzone in cerebellar cortical lobule VI (HVI) have shown changes in activity consistent with encoding the conditioned behavioural response (Jirenhed et al., 2007). However, previous studies of cerebellar activity during voluntary, multijoint limb movement such as reaching (e.g. Fortier et al., 1989; Greger et al., 2004; Hewitt et al., 2011; Miles et al., 2006; Miller et al., 2002) have not considered activity patterns in relation to microzonal organization. The question therefore remains whether there are systematic differences in patterns of activity between neurons located in different individual microzones that relate to different aspects of a voluntary movement, or whether all microzones within a given zone display the same general pattern of activity.

To address this we have characterized, by means of electrophysiological recording techniques, receptive fields of single units in the forelimb part of the C3 zone in awake cats and analysed their patterns of activity during the performance of a visually guided forelimb reaching task. Our experiments demonstrate that in the C3 zone of the awake animal: (i) the spatial organization of receptive fields of different neuronal elements and the topographical organization of their localization is consistent with the microzonal arrangement described in the anaesthetized animal; (ii) individually, the neuronal elements vary in their patterns of activity during performance of a reaching task; and (iii) Purkinje cells with different receptive fields, and therefore belonging to different microzones, together encode different aspects of the broader motor function controlled by the whole cerebellar zone. Our results are therefore consistent with the function of the C3 zone during voluntary behaviour being organized within a microzonal frame of reference.

## Material and methods

### Ethical approval

All experimental and surgical procedures were performed in accordance with the UK Animals (Scientific Procedures) Act 1986 and were approved by the University of Bristol Animal Welfare and Ethical Review Body (PPL 30/3058). The investigators understand the ethical principles under which the *Journal of Physiology* operates and the work within this study fully complies with the journal's animal ethics checklist. All steps were taken to minimize pain and suffering.

### Behavioural training

Nine purpose-bred adult male cats (4–6.5 kg, bred in-house except one animal supplied by Isoquimen, SL, Barcelona, Spain) were trained for *ca.* 6–8 weeks to perform a reaching task. No aversive training techniques were used. Food rewards were given during training and recording sessions that were over and above the *ad libitum* chow given in the home pen. Cats enthusiastically welcomed the training/recording sessions and entered the reaching set up without any coaxing. Prior to training in the reach task, time was spent each day ensuring that the animals were used to being patted, tapped and stroked to habituate the animals to somatosensory testing.

The behavioural task was similar to that described previously (Alstermark et al., 1986; Apps et al., 1997). Briefly, the target for reach consisted of a Perspex tube (30 mm wide) placed in front of the animal at approximately shoulder height (Fig. 1A). The tube contained a small morsel of fish as a reward for successful completion of individual reaches. The cue to reach with the left forepaw was the opening of a door in front of the tube entrance. Throughout the sessions the animals were sitting in a natural seated posture and lightly restrained by a loosely applied harness. The head was unrestrained, which meant that the animal's posture and movement were as natural as possible. At no point did any of the animals show any signs of discomfort. At the end of the recording session the animal was returned to a large communal pen where additional food was freely available.

### General surgical procedures

Following behavioural training, implantations were carried out using aseptic techniques and full surgical

anaesthesia. Animals were first sedated with an intramuscular injection of medetomidine hydrochloride (Domitor, 15 $\mu$g/kg, Vetoquinol, Towcester, UK) and the trachea intubated. Anaesthesia was induced with intravenous propofol at a rate of 0.1–0.3 mg/kg/min (PropoFlo Plus, 10 mg/ml, Zoetis, Leatherhead, UK) and maintained on gaseous isoflurane (Boehringer Ingelheim, Berkshire, UK) in oxygen for the duration of the surgical procedure, except during periods of electrophysiological investigations during which anaesthesia was transferred back to propofol for cerebellar cortical mapping (see below). The concentration of isoflurane (1.5–3%) was adjusted during surgery in response to capillary $O_2$ saturation, end tidal $CO_2$, cardiac and respiratory rates and limb withdrawal reflex to pressure applied to the paws. Body temperature was monitored

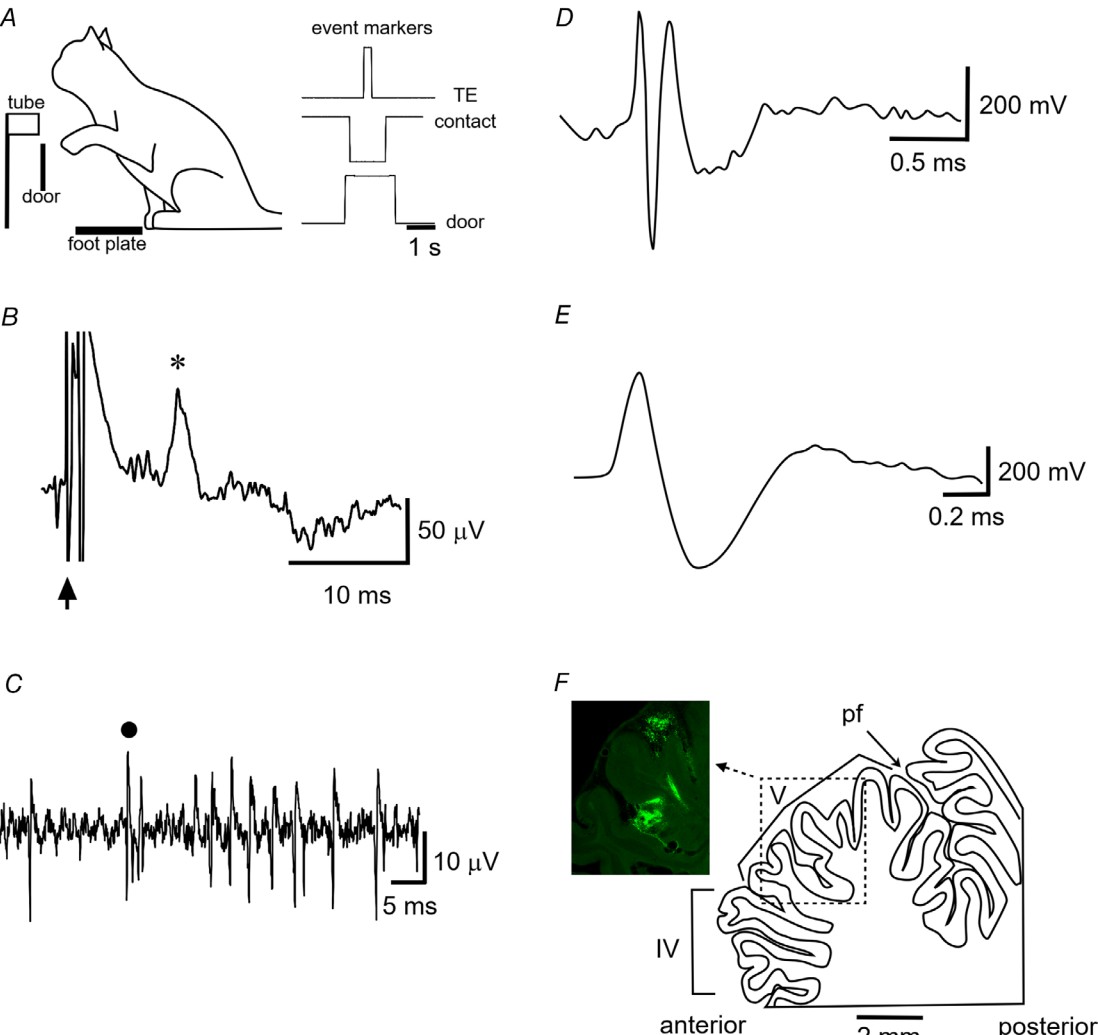

**Figure 1. Experimental arrangements**
*A*, schematic of the reach retrieval task. The cue to reach was a door covering the tube dropping down. A contact foot plate monitored paw lift-off. At the end of the reach, the paw enters the tube, thereby breaking an IR beam located at the mouth of the tube. The sequence of event markers recorded during a single reach retrieval is shown on the right. Door: upward deflection of trace represents tube door opening; Contact: downward deflection represents paw lift-off from contact plate whereas upwards deflection represents paw touch-down after retrieval of a food reward in the tube; TE: tube entry, upward deflection of paw entering the tube. Scale bar, 0.5 s. *B*, C3 zone climbing fibre field potential (asterisk) evoked by ipsilateral superficial radial nerve stimulation (onset of stimulus at time of arrow, intensity, 2T). Scale bars, 50 $\mu$V and 10 ms. *C*, example extracellular recording of a Purkinje cell in the C3 zone. Complex spike waveform marked by a dot. Scale bars, 50 $\mu$V and 5 ms *D*, example extracellular recording of a mossy fibre unit recorded in the C3 zone. Scale bars 200 mV and 0.5 ms *E*, example extracellular recording of a putative Golgi cell unit Scale bars, 200 mV and 0.2 ms. *F*, sagittal outline of cerebellar section. Dotted rectangle shows area related to photomicrograph with fluorescent material in lobule V to mark the location where the majority of recordings where made. Pf, primary fissure; V, lobule V. Scale bar, 2 mm. [Colour figure can be viewed at wileyonlinelibrary.com]

and maintained within physiological limits with the aid of a homeothermic heating blanket. At the end of surgery, a prophylactic injection of antibiotics (cefuroxime, Zinacef, 10 mg/kg, GlaxoSmithKline, London, UK) was administered subcutaneously. Following surgery, the animals were monitored closely until they had recovered full consciousness. Analgesia was administered subcutaneously with buprenorphine (Buprecare, 20 mg/kg, Animal Care, York, UK) at the end of surgery followed by meloxicam (Metacam, 0.05 mg/kg, Boehringer Ingelheim, Bracknell, UK) for ∼4–5 days after surgery. Post-surgical care included daily monitoring by the investigators and technical support staff to ensure all animals were in good health and well-being.

### Implantation procedure

Under general anaesthesia a craniotomy was made to expose the left anterior lobe paravermal cerebellum over lobule V, and a small recording chamber placed over the folia where the largest C3 zone-related electrophysiological responses were evoked by ipsilateral forelimb stimulation (see Cerebellar cortical mapping under general anaesthesia). Up to three stainless-steel T-bolts were implanted to provide additional anchorage for the chamber, with one T-bolt used served as a reference for differential recording. For bipolar nerve stimulation, two pairs of Teflon-insulated wire were implanted in the forelimbs around each of the left and right superficial radial nerves. A unipolar lead was implanted subcutaneously into the distal left forelimb to detect paw lift during reaching. Another lead was implanted in the back and served as an earth. All leads were fed subcutaneously from head-mounted micro-D connectors that were attached to the skull with dental acrylic cement, and a headpiece fashioned so as to incorporate the chamber.

### Cerebellar cortical mapping under general anaesthesia

Under general anaesthesia, pairs of percutaneous stimulating needles were inserted temporarily into the ipsilateral and contralateral forelimbs and one or two brief (0.1 ms, 1 kHz) square wave electrical pulses were delivered at an intensity to evoke a visible twitch in the stimulated limb. The responses to peripheral stimulation were recorded on the cerebellar surface using a lightly sprung ball electrode in order to identify the C zones of lobule V electrophysiologically. The resulting field potentials were amplified and bandpass filtered (0.03–5 kHz). Responses were classified according to previously defined response latencies and the presence or absence of a contralateral response. Evoked fields in the C2

zone were defined as having a latency longer than 17 ms in response to both ipsi- and contralateral stimulation whereas the neighbouring C1 and C3 zones were defined as having shorter (10–16 ms) response latencies evoked by ipsilateral limb stimulation only (Apps & Garwicz, 2000; Ekerot & Larson, 1979; Trott & Armstrong, 1987). The recording chamber (see Implantation procedure) was placed over the centre of the electrophysiologically defined C3 zone, where the largest cerebellar responses were evoked by peripheral stimulation of the ipsilateral forelimb.

### Recording arrangements and data acquisition in awake animals

Extracellular single-unit recordings confined to superficial cerebellar cortical layers were obtained in the awake animal during the reaching task using either parylene-insulated tungsten microelectrodes (impedance 2–5 MΩ; WPI, Sarasota, FL, USA) or a 16-channel Vector Array microelectrode (Neuronexus, Ann Arbor, MI, USA). Recording sessions were carried out over a period of ∼8 weeks (range 4–12 weeks), 6 days per week. At the start of the recording session, the animal's head was lightly held for a duration of ∼5 s by an experimenter whilst another experimenter placed and secured the microdrive onto the chamber. Once the microdrive was secured, the animal was free to move its head. The microelectrodes were advanced using a stepping microdrive (NAN Instruments, Nof Hagalil, Israel) mounted onto an $x$–$y$ micromanipulator stage that was attached to the recording chamber during recording sessions. The $x$–$y$ manipulator stage allowed the spatial coordinates of each electrode track to be precisely adjusted relative to the centre of the chamber. Signals were amplified and bandpass filtered (0.3–7.5 kHz). Paw lift-off from the foot plate and paw entry into the tube were continuously monitored via a high-frequency carrier signal applied to a copper contact plate and an IR beam located at the mouth of the tube, respectively (Fig. 1*A*). The opening of the door to the tube (the cue for the animal to reach) was monitored via a microswitch.

All neural and task event signals were captured using either customized Spike2 software running on a CED 1401Plus computer interface unit (Cambridge Electronic Design, Cambridge, UK) or a Cerebus Neural Signal Processor (Blackrock Microsystems LLC, Salt Lake City, UT, USA).

### Experimental design

At the start of each recording session in the awake animal, we confirmed that the zonal location of the microelectrode was in the C3 zone from the latency of the climbing

fibre field potentials evoked by electrical stimulation of the ipsilateral and contralateral superficial radial nerves (Fig. 1*B*). Low-intensity (non-noxious) pulse stimulation was used, with the intensity increased until a small but visible limb twitch was observed. This was typically at an intensity twice the threshold (2T) for detecting a cerebellar cortical field potential. Zonal identity was determined as described above.

Whilst the animal was sitting quietly, the micro-electrode was advanced into the cerebellar cortex in small steps until a unit was isolated. The animals showed no signs of distress or discomfort at any point in the procedure. Single units were identified by electro-physiological features of spontaneous and evoked neuronal activity and depth below the cerebellar surface. To ensure that we remained within the cortical zone identified at the start of the recording session, recordings were confined to the most superficial layers of the cerebellar cortex.

Purkinje cells were identified by the presence of complex spikes (Fig. 1*C*; Armstrong & Rawson, 1979; Bloedel & Roberts, 1971). In accordance with previous studies (Garwicz & Andersson, 1992; van Kan et al., 1993) units that displayed a triphasic waveform and wide dynamic firing range, varying from silence to high-frequency bursts at several hundred impulses per second, were classified as mossy fibres (see Fig. 1*D*). Putative Golgi cells were identified by their distinctive regular and low-frequency discharge, interspike interval histograms with broad peaks, long spike durations (see Fig. 1*E*) and large tuning distance (Edgley & Lidierth,

1987). Units that could not be categorized as any of the above were classified as putative cortical interneurons.

Once a unit was isolated and identified, the spatial organization of the receptive field for any peripheral somatic afferent inputs was determined and delineated by carefully delivering manual stimuli such as brushing of hairs, tapping of skin, palpation of muscle or passive movement of joints. No noxious stimuli were used. As the animals were very well habituated to somatosensory testing, they remained passive during receptive field mapping and did not show any behavioural responses (i.e. they would sit quietly during testing). Receptive field classification was based on previous detailed studies of the C3 zone in the anaesthetized cat (Ekerot & Jorntell, 2001; Ekerot et al., 1991a; Garwicz et al., 1998). The microzonal organization of the C3 zone was originally based on classification of climbing fibre receptive fields (Ekerot et al., 1991a), but was extended to the mossy fibre system (Garwicz et al., 1998) and comprises a total of 10 classes and approximately 40 subclasses. In the present study we used this classification to group single units with similar receptive fields to determine the topographical organization of input to the C3 zone. Receptive field identity was verified by at least two experimenters (and in many cases three, MG, DMH and NLC) who would reach consensus about the receptive field class. Topographical maps of receptive field identity as a function of spatial location within the zone were constructed using *x–y* coordinates of the micromanipulator stage (Fig. 2).

After receptive field mapping was complete, the animal began the task (described above) until they reached satiety.

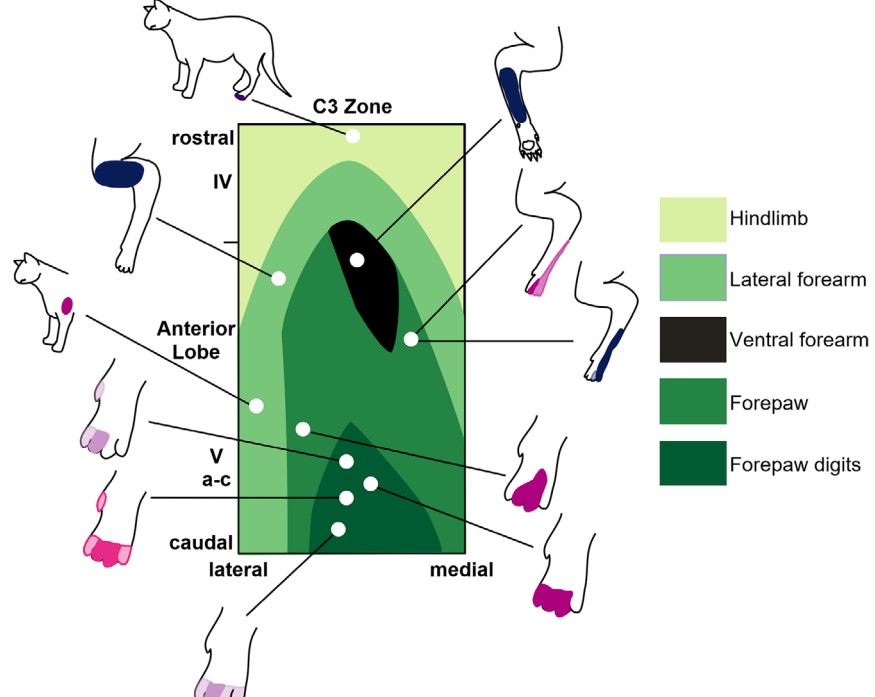

**Figure 2. Topographic organization of the C3 zone in the awake cat**
Map showing limb receptive field locations within the C3 zone in the anterior lobe as determined in the anaesthetized cat (Ekerot et al., 1991a). Cerebellar lobules are indicated to the left. Figurines surrounding the map show example peripheral receptive fields in the awake cat recorded in different recording sessions from two animals. Figurines with receptive fields in dark blue are Purkinje cell complex spikes, in dark purple are Purkinje cell simple spikes, in light purple mossy fibres and pink cortical interneurons. Darker shading in figurines denotes regions of skin that generated the strongest response within the receptive field; lighter areas denote the total extent of the receptive field. [Colour figure can be viewed at wileyonlinelibrary.com]

Alternatively, for approximately half of the units, we began with the reaching task and then mapped the receptive field (as described above) once we had accumulated sufficient trials or when the animal had reached satiety. This allowed us to record from a sufficient population of units with an identified receptive field class and obtain enough trials of their activity during the behaviour for quantitative analysis. In some instances, the task was stopped midway and receptive field location checked to ensure the receptive field location remained stable. Overall, our careful and conservative approach when testing and classifying the units means that we are confident that receptive field maps were accurate, and that receptive field testing was unaffected by behavioural responses or alterations in the strength of the stimuli delivered.

Following a period of productive single-unit recording, in one animal, a 22-gauge bevelled needle was affixed to the microdrive tower in place of a recording electrode. The animal's head was lightly held for a duration of ~5 s by an experimenter whilst another experimenter placed and secured the microdrive onto the chamber. Once the microdrive was secured, the animal was free to move its head. The microdive was advanced to a depth approximately equal to that used during previous recordings (~2 mm from cortical surface). The animal showed no signs of distress or discomfort as the needle was being advanced. A 6 $\mu$l dose of lidocaine (Lignol, Dechra Veterinary Products, Shrewsbury, UK) was infused 1 $\mu$l/min by means of an infusion pump (Harvard Apparatus, Holliston, MA, USA) connected to the needle via surgical tubing. The animal was then allowed to proceed on the reaching task.

### Spike sorting

Spikes were sorted with standard spike-sorting procedures off-line using either a clustering algorithm based on template matching and principal component analysis within Spike2 or with a semiautomated clustering algorithm (KlustaKwik; Kadir et al., 2014). The resulting clusters were manually inspected and refined with MClust (A. D. Redish and colleagues; https://github.com/adredish/MClust-Spike-Sorting-Toolbox). Isolation of single units was verified by the presence of a distinct refractory period in the interspike interval (ISI) histograms.

### Data analysis

Spike times and behavioural events (paw lift, tube entry) were analysed with custom-written scripts in MATLAB (v. 2018a). For each unit, perievent-time histograms (PETHs) of spike firing rates (bin width 10 ms) were aligned to the onset of paw lift and convolved with a Gaussian kernel

of 10 ms. Our analysis was confined to paw lift as the number of units that responded to other events (e.g. grasp, paw down) was low. To compare firing patterns across units, spike activity was transformed to $z$-scores using the baseline mean firing, where baseline was $-1000$ to $-500$ ms prior to the onset of movement. This baseline period avoided any preparatory changes in activity prior to movement. Changes in movement-related activity were considered significant when firing rates crossed $\pm2$ SD of the baseline mean. Onset latency was calculated to the leading edge of the first significant bin.

For Purkinje cells where both simple spike and complex spike receptive field maps could be reliably determined, the area of overlap was estimated by superimposition of maps. Area of overlap was expressed as the percentage of the total extent of the two receptive fields (see each pair of black and grey shaded areas in Fig. 4 combined).

The duration of the reach was defined as the time interval between the paw lifting off the contact plate and entering the Perspex tube. Reach durations were used as a proxy to describe the reaching behaviour. Individual reaches were excluded from further analysis if they displayed reaction times of less than 150 ms to the door opening, and/or if they displayed reach durations greater than 2.5 SD from the mean (Miller, 1991). The proportion of trials excluded based on these criteria was 12.5%.

We used a cluster-based method to determine whether units belonging to the same receptive field class had similar activity patterns. Principal component analysis (PCA) was used to reduce noise; the standardised $z$-scored firing activity for each unit in relation to paw lift with a time duration of 2 s was treated as a multidimensional vector with each time point regarded as a component. This dimensionality was reduced using PCA: the first three principal components together explained $> 80\%$ of the variance of the data (Forgy, 1965); the individual $z$-scored firing activity profiles were projected onto these three components and the resulting noise-reduced profiles were then used to calculate Euclidean distances between the units.

These data were then clustered using $k$-means with the number of clusters chosen to match the number of available receptive field classes. This clustering grouped together units with similar activity patterns. A confusion matrix was used to describe the relationship between this $k$-means clustering of the data and grouping by receptive field class. To quantify how well the $k$-means clusters and receptive field class groups matched, we calculated the Adjusted Rand Index (ARI) (Hubert & Arabie, 1985; Rand, 1971); since the $k$-means clustering reflected the neuronal pattern of activity, this index measured the correspondence between activity pattern and receptive field class. To convert the ARI into a statistic, the index was also calculated for simulated confusion matrices in which

the receptive field class groups were assigned randomly; the values for 1000 of these simulated confusion matrices were compared to the actual ARI for the real confusion matrix. The closer the value of this statistic is to 100% the less likely any correspondence between receptive field class and activity patterns is by chance.

### Histology

At the end of each experiment, animals were humanely killed using Home Office approved methods. Animals were deeply anaesthetized with an intraperitoneal injection of pentobarbital (Dolethal, Vetoquinol, Towcester, UK). In two cases, 200–400 nl of fluorescent-tagged material (20% solution of Fluoro-Emerald; Molecular Probes, Eugene, OR, USA, combined with green beads; Lumafluor, New City, NY, USA) was injected into the cerebellar cortex (0.5–1 mm below the surface of the floor of the recording chamber) to mark the location where the majority of recordings where made (Fig. 1*F*). All animals were perfused transcardially with isotonic saline followed by 4% paraformaldehyde. The cerebellum was removed and stored in 4% paraformaldehyde at 4°C. The brain was transferred to a 30% sucrose solution after a week and allowed to sink. A freezing microtome was used to cut the cerebellum into 50 $\mu$m sagittal sections. Images were acquired using a Zeiss Axioskop II inverted microscope equipped with a CooLED pE-100 excitation system and Ocular digital monochrome camera and software.

### Statistics

Data are expressed as mean $\pm$ SD unless stated otherwise. Statistical analyses were conducted using SPSS (IBM, Armonk, NY, USA). Normality was assessed using a Shapiro-Wilk test. Parametric and non-parametric tests were used accordingly. Statistical significance was set at $P < 0.05$.

### Results

### General features of the cerebellar neuronal population

In nine cats, extracellular recordings were obtained from a total of 208 single units located in cerebellar lobules IV and V (Fig. 1*D*) within a region of paravermal cortex identified electrophysiologically as the forelimb part of the C3 zone (see Fig. 1*B*). We focused our analysis on single units with a clearly defined peripheral receptive field located on the ipsilateral forelimb or shoulder (see Methods). This allowed direct comparison with previous characterization of forelimb-related C3 microzones in the anaesthetized cat (Ekerot et al., 1991a; Garwicz et al.,

1998). A total of 65 single units met this strict criterion. Of the remaining 145 units, 58 units were termed 'partly classified' because they had a receptive field on the ipsilateral forelimb, but a more precise classification was not possible because the unit was lost during mapping or during task performance. A further 60 units had a receptive field located on the face, head or neck and three units had receptive fields on the ipsilateral hindlimb. These units were all C3 in origin but were not investigated further because the microzonal organization has not been mapped previously for these areas of the body. A further 13 units were lost before we were able to discern their peripheral receptive field location. Finally, seven units were labelled 'unclassified' as, despite extensive testing, we were unable to determine a peripheral receptive field. These unclassified cells represented just 3.4% of the total population tested ($n = 208$ units). Therefore, the large majority of units in our sample were associated with a peripheral somatic receptive field.

Of the 65 units included in our primary analysis, 46 were classified as Purkinje cells (of which 17 units had peripheral receptive fields defined for complex spike activity and 38 units had receptive fields defined for simple spike activity, indicating that for nine Purkinje cells both complex spike and simple spike receptive fields were characterized, see Fig. 1*C*), 13 units were classified as mossy fibres and six units as putative cortical interneurons (PCIs, four Golgi cells, two molecular layer interneurons; see Methods for physiological identification of different cell types). Most units were found to be activated either by brushing of hairs or light tapping of the skin of the ipsilateral forelimb. Some units required pressure and palpation of deeper tissue within the forelimb, although none were found to be activated by palpation of individual muscles.

### Topographical organization of C3 microzones in the awake animal

The topographical organization of Purkinje cell climbing fibre receptive fields within the C3 zone in cerebellar lobules IV and V, as determined previously in the anaesthetized cat, is shown in Fig. 2. As a first step, it was important to establish whether this arrangement was similar in the awake animal. In the awake cat, Purkinje cells with climbing fibre receptive fields on proximal and ventral areas of the ipsilateral forelimb were located in rostral folia of lobule V, while receptive fields restricted to the ipsilateral hindlimb were located further rostrally (figurines with receptive fields shaded dark blue in Fig. 2). The peripheral receptive fields of units identified as mossy fibres, PCIs or the simple spike activity of Purkinje cells could also be fitted to a similar spatial organization (figurines with receptive fields shaded

light purple, pink or dark purple, respectively in Fig. 2). Therefore, regardless of neuronal type, the topographical organization of cerebellar cortical units located within the C3 zone in the awake cat was consistent with the detailed mapping previously reported in the anaesthetized animal and, as such, the original classification system (Ekerot et al., 1991a; Garwicz et al., 1998) was used as our framework for categorizing all unit types.

### Forelimb receptive field classification

As shown in Fig. 3, the receptive field mapping was usually sufficiently detailed to allow a characterization of units at the resolution of subclasses, according to the scheme previously developed in anaesthetized animals, and hence, with respect to topography, at the level of individual microzones. However, because of the practical constraints associated with single unit yields in the awake, lightly restrained (non-head-fixed) animal, receptive field categorization was confined to classes. Receptive field data were available for all but one of the 10 receptive field classes previously described in the anaesthetized animal (classes 1–6, 8–10). Figure 3 shows for each of these classes the total area of the receptive fields found for each neuronal type (Purkinje cell complex spikes, Purkinje cell simple spikes, mossy fibres and PCIs). Overall, the receptive fields covered most parts of the ipsilateral forelimb, apart

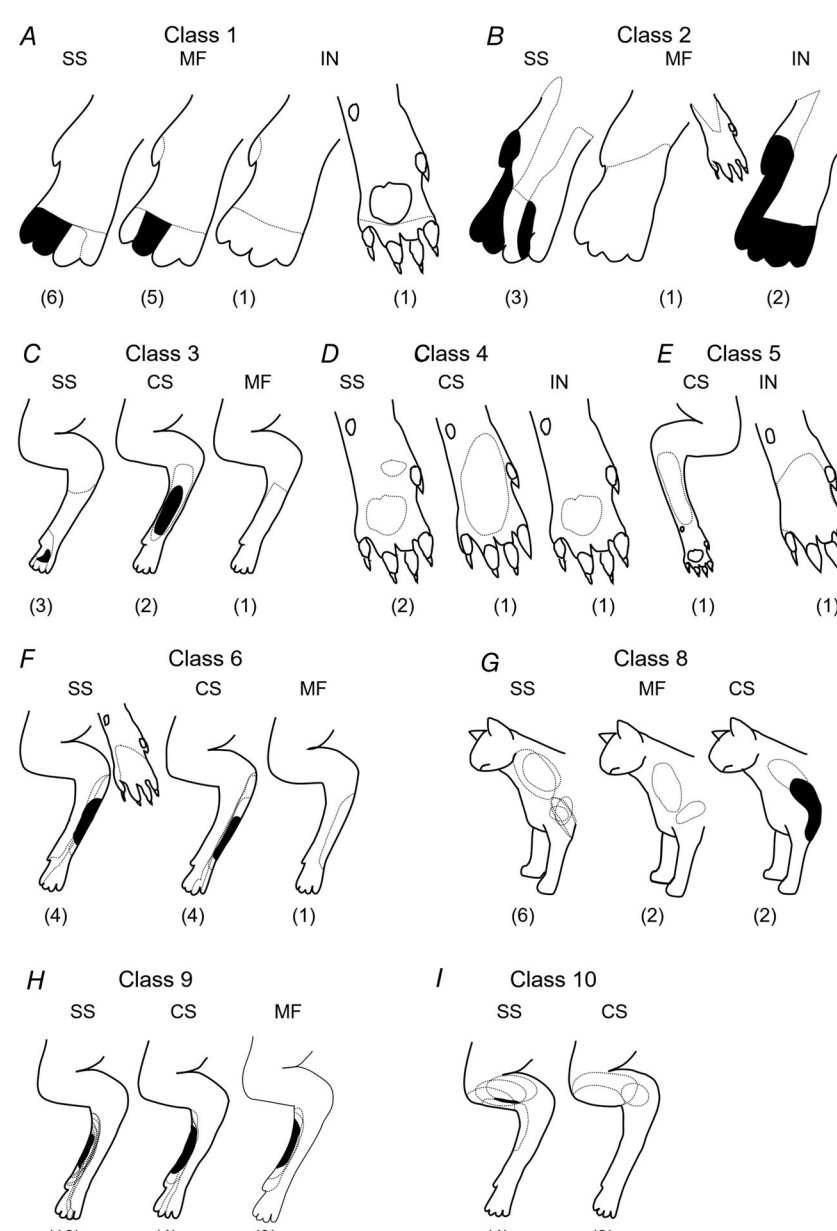

**Figure 3. Classification of cerebellar cortical units according to receptive field class**

Units were categorized in accordance with previous classification of Purkinje cell complex spike (Ekerot et al., 1991a) and mossy fibre receptive field characteristics (Garwicz et al., 1998). Each figurine shows all available receptive fields superimposed for each class and subdivided into those units identified as Purkinje cell simple spikes (SS), complex spikes (CS), mossy fibres (MF) or putative cortical interneurons (IN). Numbers in parentheses below each figurine indicate the number of units per type. *A*, Class 1; *B*, Class 2; *C*, Class 3; *D*, Class 4; *E*, Class 5; *F*, Class 6; *G*, Class 8; *H*, Class 9; *I*, Class 10. Dashed lines indicate the total extent of single receptive fields while areas filled in black indicate where overlap occurred in every available case.

from the ventral surface of the upper forelimb. Class 1 units were those with the most distal receptive fields: centred on the ventral and dorsal surface of the digits and proximal borders close to the metacarpophalangeal joints (Fig. 3*A*, *n* = 13 units). Class 2 units were those with the receptive field centred (Fig. 3*B*, *n* = 6 units) on the ventral and dorsal surfaces of the digits and the forepaw and proximal borders close to the wrist, whereas class 3 units (Fig. 3*C*, *n* = 6 units) had receptive fields that were located predominantly on the dorsal paw and forearm. Class 4 units (Fig. 3*D*, *n* = 4 units) were those to receptive fields located predominantly on the ventral surface of the paw. Class 5 (Fig. 3*E*, *n* = 2 units) receptive fields were also confined to the ventral surface of the limb, but with more proximal centres, on the paw and wrist or forearm, while receptive fields located on the lateral part of the forearm were identified as class 6 (Fig. 3*F*, *n* = 9 units). No cells were found with receptive field characteristics consistent with class 7 (i.e. lateral forearm and upper arm receptive fields). Class 8 cells were those with receptive fields located on the lateral shoulder/upper arm and neck (Fig. 3*G*, *n* = 10), whereas units with receptive fields located on the radial/medial part of the paw and forearm were defined as class 9 (Fig. 3*H*, *n* = 17 units). Finally, class 10 cells (Fig. 3*I*, *n* = 7 units) were those with receptive fields mainly confined to the medial shoulder/upper arm (Ekerot et al., 1991a; Garwicz et al., 1998; Jorntell et al., 1996).

## Comparison of receptive fields of simple spikes and complex spikes from the same Purkinje cells

An important key to understanding how information is processed in the cerebellar cortex is the pattern of convergence and interaction between climbing fibre and mossy fibre input at the level of individual Purkinje cells. Climbing fibres generate complex spikes due to direct termination on Purkinje cells, while mossy fibres generate simple spikes indirectly via the granule cell–parallel fibre pathway. The receptive fields of Purkinje cell complex spikes are generally similar to those of subjacent granule cells and mossy fibres (e.g. Ekerot & Larson, 1980; Garwicz et al., 1998; Gerrits et al., 1985; Pijpers et al., 2006; see also Shimuta et al., 2020). However, opinion is divided on whether Purkinje cell simple spike receptive fields are due mainly to mossy fibre influence on local granule cells (Bower & Woolston, 1983; Brown & Bower, 2001; Cohen & Yarom, 1998; Isope & Barbour, 2002), or non-local granule cells (Dean et al., 2010; Ekerot & Jorntell, 2001; Ekerot & Jorntell, 2003; Jorntell & Ekerot, 2002).

For nine Purkinje cells we were able to reliably discriminate simple spikes and complex spikes from the same units and define their corresponding receptive fields in the awake animal (Fig. 4). Receptive fields of complex

spikes and simple spikes in six Purkinje cells (67%) were found to have similar peripheral receptive fields. Maximum overlap between complex spike and simple spike receptive fields was on average 64.4 ± 35.6 (*n* = 6, pairs of receptive fields outlined in green in Fig. 4). The remaining three cells differed markedly in the location of their complex spike and simple spike receptive fields

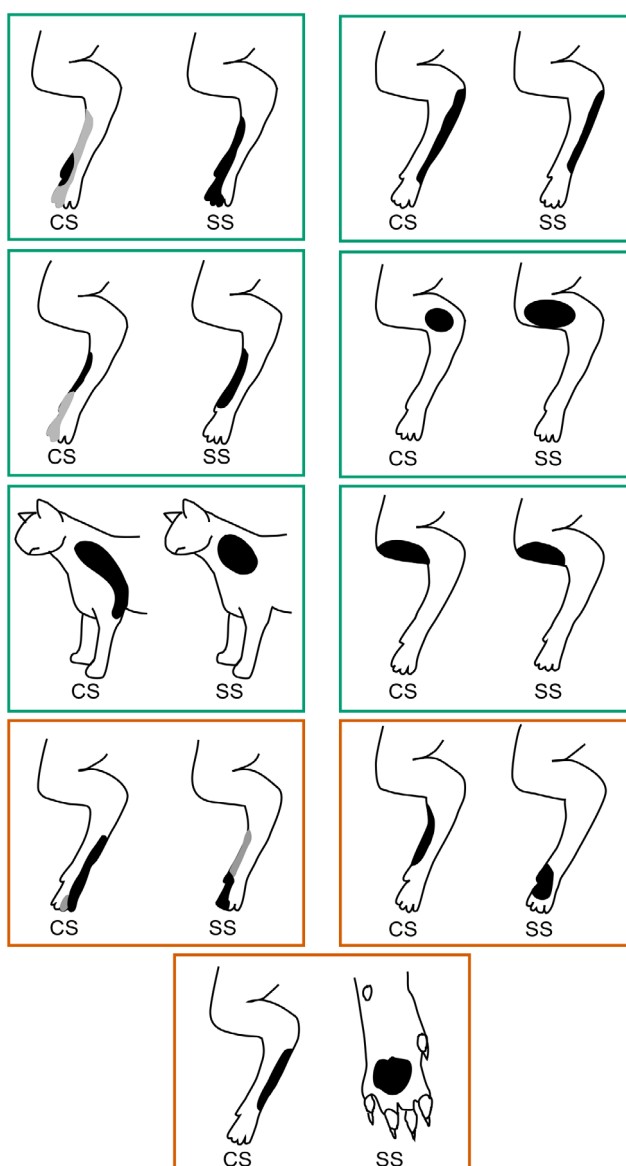

**Figure 4. Receptive fields of complex spikes and simple spikes recorded from the same Purkinje cells**
Complex spike and simple spike receptive fields for individual Purkinje cells (*n* = 9) recorded when the animal was at rest. Pairs of receptive fields with the same receptive field classification are enclosed by green boxes; pairs with different receptive field classes are enclosed by red boxes. On each figurine black shading denotes regions of skin that generated the strongest responses, and grey indicates the total extent of the receptive field. CS, complex spike; SS, simple spike. [Colour figure can be viewed at wileyonlinelibrary.com]

which, on average, had a maximum overlap of only $2.2 \pm 3.8$ ($n = 3$, pairs of receptive fields outlined in red in Fig. 4). Thus, in our sample of Purkinje cells the findings were mixed: for the majority the overlap in receptive fields is consistent with the hypothesis that the modulation of simple spike activity by sensory inputs occurs as a consequence of excitation of local granule cells, whereas for the remaining Purkinje cells the results are consistent with excitation from granule cells located non-locally. However, clearly caution is needed in interpreting a small sample size, and further studies will be required to test these findings more fully.

### Task performance and pattern of activity of units within different receptive field classes

The target for reach consisted of a Perspex tube containing a food reward and was located at approximately shoulder height to the animal (Fig. 1*A*). The cue to reach with the left forepaw was the opening of a door in front of the tube entrance. Consistent with the C3 zone being critically involved in this voluntary, goal-directed behaviour, infusion of lidocaine in one animal caused a marked slowing of reaching to the target.

Of the 65 units grouped by class, 43 units [28 simple spikes, six complex spikes, four PCIs (four putative Golgi cells) and nine mossy fibres] were recorded during task performance for a sufficient number of trials to permit quantitative analysis of neuronal activity during reaching, and also to examine the relationship between patterns of response and receptive field characterization. The activity of almost all available units was clearly modulated at some point in the task, and the great majority exhibited strong modulations (a change in firing rate $\geq$ 2SD from baseline, see below). The largest peak of modulation was often accompanied by secondary, more subtle but distinct variations in discharge rate at other times in the task. This probably reflected the fact that the multijoint limb movement consisted of several key elements: (i) paw lift and forelimb reach, (ii) grasp, (iii) retrieval (including chewing and postural adjustments) and (iv) paw down.

The complexity of the pattern of task-related modulation was broadly related to location of the peripheral receptive field. Units with distal (digit, paw, wrist and elbow receptive fields, classes 1–6) tended to show the most complex patterns of modulation (i.e. tended to have two or three peaks of activity; Fig. 5*A*). They often changed their discharge rate before the time of paw lift or paw down, and also between the times of entry to and exit from the tube (when the fish morsel was grasped). By comparison, units with proximal or axial receptive fields (shoulder, neck, classes 8–10) tended to show simpler activity profiles (i.e. tended to have only one peak in activity; Fig. 5*B*).

Further analysis was confined to classes 1, 2, 6, 8, 9 and 10 (as each of these classes had four or more units), and also activity around the time of paw lift because this was a temporally precise task-related event and most units displayed modulation of their activity around this time. Given the anatomical and physiological framework of C3 microzones we grouped different types of unit together based on whether they belonged to the same receptive field class. Figure 6A–F shows representative examples of the patterns of activity of units from classes 1, 2, 6, 8, 9 and 10 in relation to paw lift. Baseline firing rate between classes was not statistically significant (class 1 $32.4 \pm 15.5$ Hz, $n = 8$ units; class 2 $22.6 \pm 13.4$ Hz, $n = 4$ units; class 6 $37.6 \pm 7.2$ Hz, $n = 5$ units; class 8 $31.6 \pm 16.5$ Hz, $n = 7$; class 9 $33.9 \pm 30.1$ Hz $n = 10$ units; class 10 $36.7 \pm 21$ Hz, $n = 4$ units, one-way ANOVA $P = 0.954$).

To quantify and compare changes in activity across different units, we first normalized firing rates as $z$-scores and defined changes in activity as statistically significant if they were $\pm$ 2 SDs from baseline (see Methods). Most units showed a significant increase in activity around the time of paw lift (Fig. 6*G*). Peak firing rates for units that increased or decreased compared to baseline were statistically significant (increase: baseline rate $= 35.2 \pm 24.5$ Hz, peak rate $= 60 \pm 42.1$ Hz, $n = 30$, $P < 0.0001$; decrease: baseline rate $= 23.8 \pm 10.9$ Hz, peak rate $= 11.7 \pm 7.8$ Hz, $n = 8$, $P < 0.0001$, paired $t$ test, Fig. 6*H*).

On average, the activity of all available units in class 1 and class 6 exhibited modulation in firing rate preceding paw lift (onset latency class 1, $-110 \pm 63$ ms, $n = 8$, mean $\pm$ SD; class 6, $-152 \pm 83$ ms, $n = 5$). By comparison, the average activity of all units in class 2 and class 9 showed a modulation in firing rate around the time of paw lift (onset latency class 2, $0 \pm 165$ ms, $n = 4$; class 9, $-8 \pm 180$ ms, $n = 10$). Class 8 and class 10 units exhibited modulations in firing after paw lift (onset latency class 8, $41 \pm 159$ ms, $n = 7$; class 10, $180 \pm 164$ ms, $n = 4$). The differences between onset latencies were statistically significant for classes 1 and 6 compared with class 10 (one-way ANOVA, $P = 0.015$, *post hoc* Tukey's class 1 *vs.* class 10 $P = 0.025$, class 6 *vs.* class 10 $P = 0.017$, Fig. 6*I*). Peak latencies also differed between classes with those located distally (classes 1, 2 and 6) occurring sooner than those classes located proximally (classes 8−10), but this was not statistically significant (peak latency, class 1, $57.5 \pm 171$ ms, $n = 8$; class 2, $60 \pm 148$ ms, $n = 4$; class 6, $-24 \pm 64$ ms, $n = 5$; class 8, $151 \pm 90$ ms, $n = 7$; class 9, $41 \pm 152$ ms, $n = 10$; and class 10, $195 \pm 181$ ms, $n = 4$, one-way ANOVA $P = 0.122$; Fig. 6*I*).

Previous studies in cats have shown that the reaching task used in the present study is highly stereotypical, with little variation between trials in terms of movement trajectories and velocity with cats adopting a very similar kinematic reaching strategy (Alstermark et al., 1993;

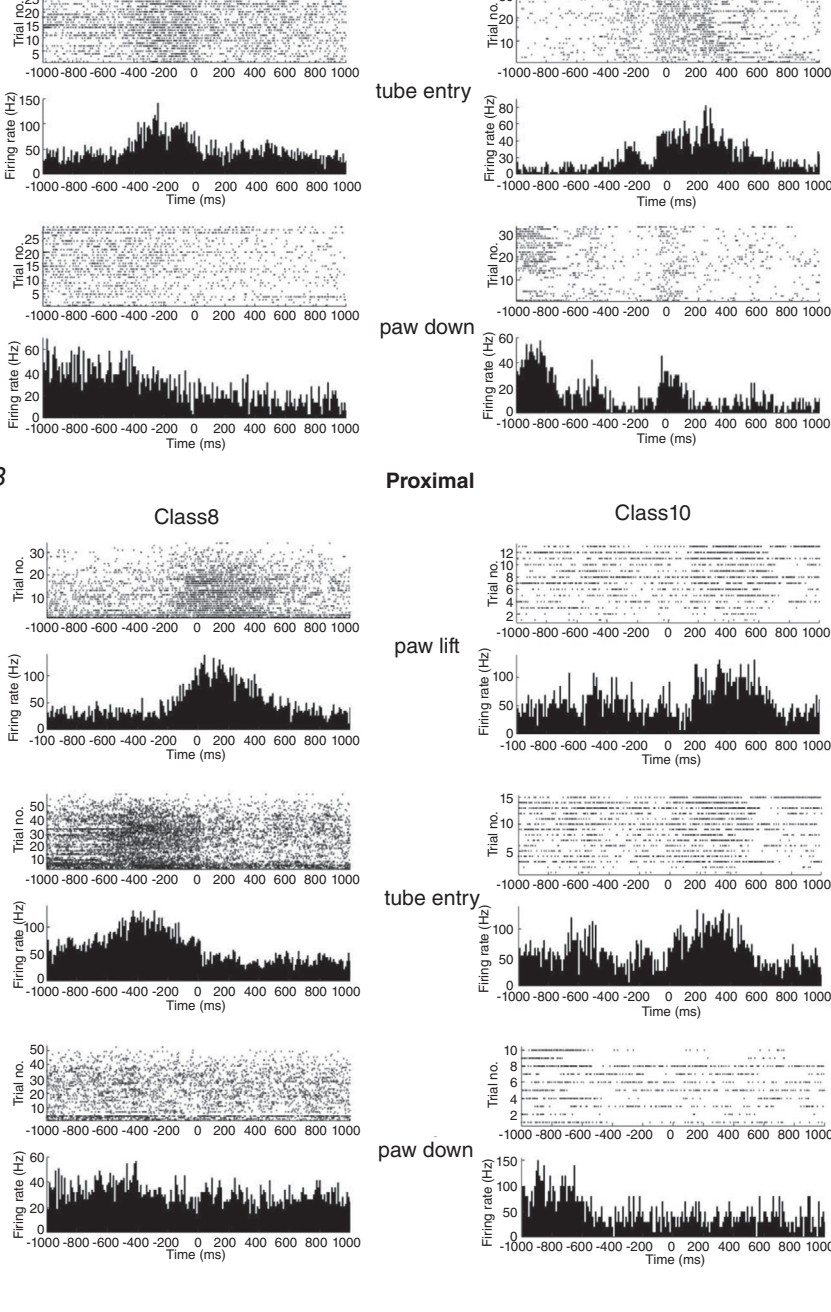

**Figure 5. Pattern of activity during reaching of units with distal *versus* proximal limb receptive fields**
*A*, example raster plots and peri-event time histograms (PETHs) for two classes of units (classes 1 and 5) with receptive fields located on the distal part of the ipsilateral forelimb. *B*, same as *A* but two example units (classes 8 and 10) with receptive fields located on the proximal part of the ipsilateral forelimb. In *A* and *B*, the top pair of panels shows rasters and PETHs with paw lift-off at time zero, middle pair of panels is tube entry at time zero and bottom pair of panels is paw touch-down at time zero.

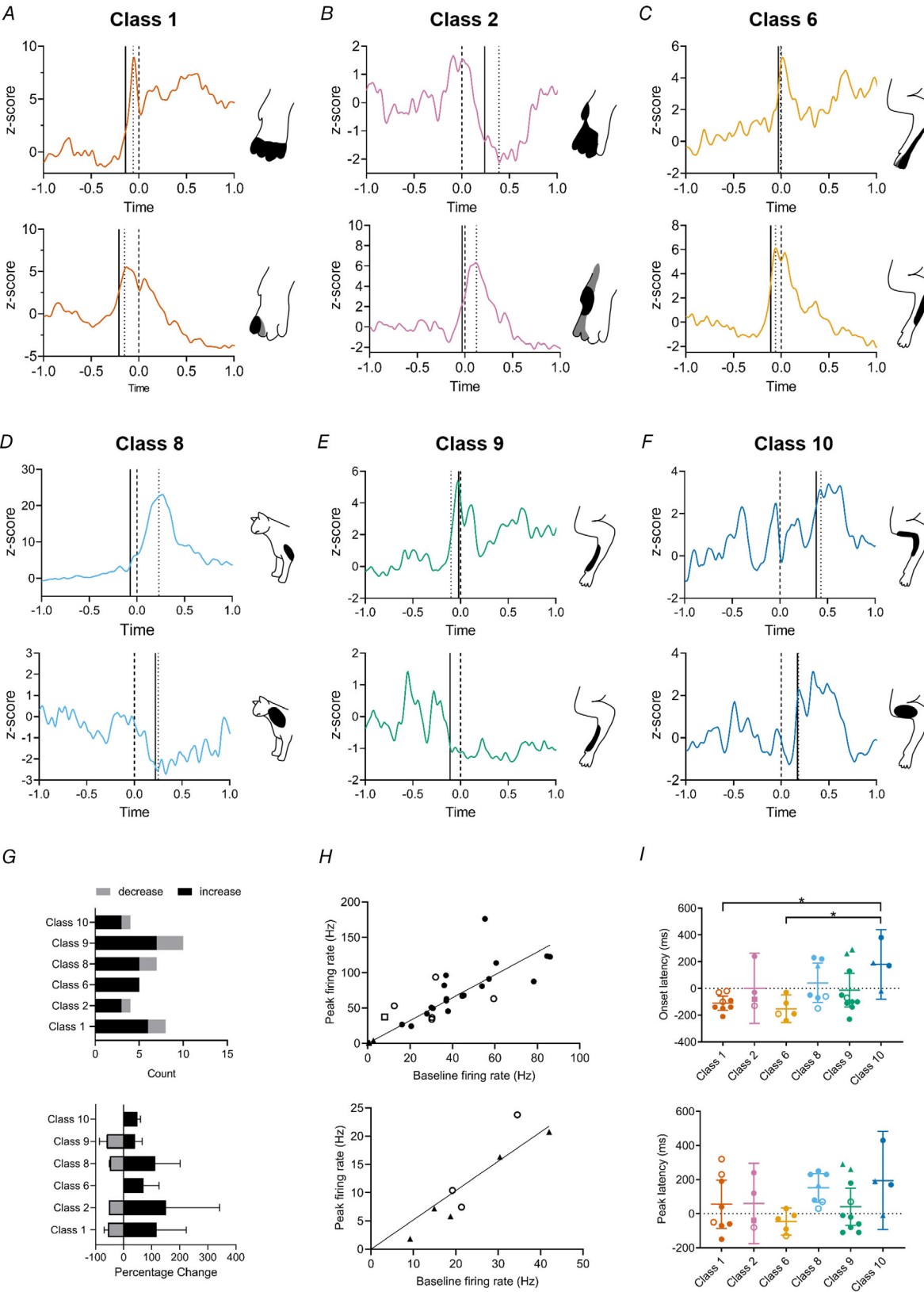

**Figure 6. Individual patterns of activity during reaching of units within the same class**

*A–F*, example of normalized PETHs aligned to paw lift (time zero, vertical dashed line; onset latency, vertical continuous line; peak latency, vertical dotted line) for 12 units from six different receptive field classes. Inset shows

receptive field location for each unit. Black shading denotes regions of skin that generated the strongest response; light grey areas denote total extent of receptive field. *G*, top panel, for each class the number of available units that displayed either an increase (black) or decrease (grey) in activity during paw lift-off. Bottom panel shows the average change in activity for the units that increased (black) or decreased (grey) their activity (mean ± SD) *H*, changes in baseline *vs*. peak firing rate for all units which either increased (top) or decreased (bottom) their activity following paw lift. Diagonal black line is line of equivalence. *I*, top panel, onset latencies of significant change in unit activity per receptive field class in relation to paw lift-off (horizontal dotted line). Bottom panel, same as left panel but peak latency. Individual data points and mean ± 95% confidence intervals. Filled circles = Purkinje cell-simple spikes; triangles = Purkinje cell-complex spikes; open circles = mossy fibres; squares = putative cortical interneurons. *$P < 0.05$. [Colour figure can be viewed at wileyonlinelibrary.com]

Schepens & Drew, 2003). In the present study analysis of reaching trajectory in one animal was consistent with this finding (see Supporting Information). However, in some animals we observed a change in reaching towards the end of the recording session when the cats were becoming satiated. When this occurred, the animals tended to slow down their movements. Whilst we generally stopped the recording session when this became apparent, the possibility remains that the difference in pattern of neural activity between units associated with different receptive field classes was related to systematic differences in reaching performance. To investigate this, we used the duration of the reach (taken from time of paw lift to entry into the tube, see Methods) as a proxy of reaching kinematics. Reach duration was on average 320 ms ± 131 ms ($n = 7$ cats, 2 animals contributed only to receptive field data). No statistical difference in reach duration was found between classes (class 1, $346 \pm 132$ msn $n = 8$; class 2, $326 \pm 126$ msn $n = 4$; class 6, $-295 \pm 115$ msn $n = 5$; class 8, $271 \pm 115$ msn $n = 7$; class 9, $318 \pm 87$ msn $n = 10$; and class 10, $251 \pm 82$ msn $n = 4$, class one-way ANOVA $P = 0.718$, $n = 6$ classes). Similarly, no statistical difference was found between class, average firing rate and reach duration [ANCOVA $F_{(5,31)} = 0.344$, $P = 0.882$, $n = 6$ classes].

We interrogated the performance of reach duration further in a sample of units ($n = 6$) where we were able to record in each case a minimum of 60 trials. For each unit we plotted the distribution of reach durations and compared fast and slow trials by considering those in the lower quartile with those in the upper quartile. Figure 7*A* shows representative data for two Purkinje cell simples spikes with PETHs derived from fast and slow trials superimposed on the same plots (black and grey PETHs respectively). Visual inspection suggests there was little difference in the pattern of neural response. Quantitative comparison of the firing rates during baseline and at peak latency (Fig. 7*B*) revealed no difference between fast and slow reaches (baseline firing rate fast = $35.1 \pm 21$ Hz, slow = $37.1 \pm 20.2$ Hz, unpaired *t* test, $P = 0.867$, $n = 6$ units; peak firing rate fast = $92.2 \pm 56.1$ Hz, slow = $81.5 \pm 51.4$ Hz, unpaired *t* test, $P = 0.738$, $n = 6$ units). Finally, Fig. 7*C* shows for each available unit the time to peak activity for the fastest trials *versus* the time to peak of slowest trials. Similar to the quantitative

analysis of firing rates, no difference was found for the latency between fast and slow trials (Pearson correlation, $r = -0.15$, $P = 0.779$). Therefore, it seems reasonable to assume that the differences in pattern of neuronal activity observed between receptive field classes was not closely related to differences in reach performance.

### Individual patterns of unit activity during reach

Inspection of the examples illustrated in Fig. 6*A–F* shows there was variation in individual unit response patterns belonging to the same receptive field class. The variation in activity in relation to paw lift across all available units is illustrated in the heat map shown in Fig. 8*A*. To quantify the extent of consistency of the pattern of modulation of individual units within each class, we used PCA combined with *k*-means clustering (Fig. 8*B* and *C*, see Methods for details). The number of clusters was based on the number of receptive field classes available for analysis with a minimum of four units within each class: in other words, *k*-means does not determine the number of clusters, but the goal here was to compare the clustering according to neuronal activity with respect to neuron classes and so *k* has been fixed so that the number of clusters and substantial classes is identical. A total of six classes met this criterion, with 4–10 units per class (class 1, $n = 8$ units; class 2, $n = 4$ units; class 6, $n = 5$ units, class 8, $n = 7$ units, class 9, $n = 10$ units; class 10, $n = 4$ units). PCA and *k*-means clustering using a shorter overall time duration found no change in the clustering.

For each unit, the comparison was visualized in a confusion matrix (Fig. 8*D*). The count in the diagonal (highlighted with dashed borders in Fig. 8*D*) shows the number of units per class in which the *k*-means clustering matched receptive field class. Classes 6 and 9 had the greatest proportion of units that matched, whereas classes 2 and 10 had none, while classes 1 and 8 were intermediate in the proportion of matched units. There appears to be confusion between classes 8, 9 and 10 and between 9 and 1. This reflects the similarity in the corresponding spike trains and could be interpreted as being the consequence of the corresponding part of the paw and shoulder being involved in similar parts of the overall movement. Classes with a greater proportion of Purkinje cells had the best fit. Clustering was therefore also performed solely on the

Purkinje cell population for classes which had four or more units (classes 1, 6, 8 and 9; see Fig. 8E–G). To estimate the performance of the confusion matrix, we calculated the ARI (see Methods for further details). The closer the ARI value to 100% the closer the fit, and a value over 95% is considered statistically significant. For the total population of available units used in Fig. 8*B*,

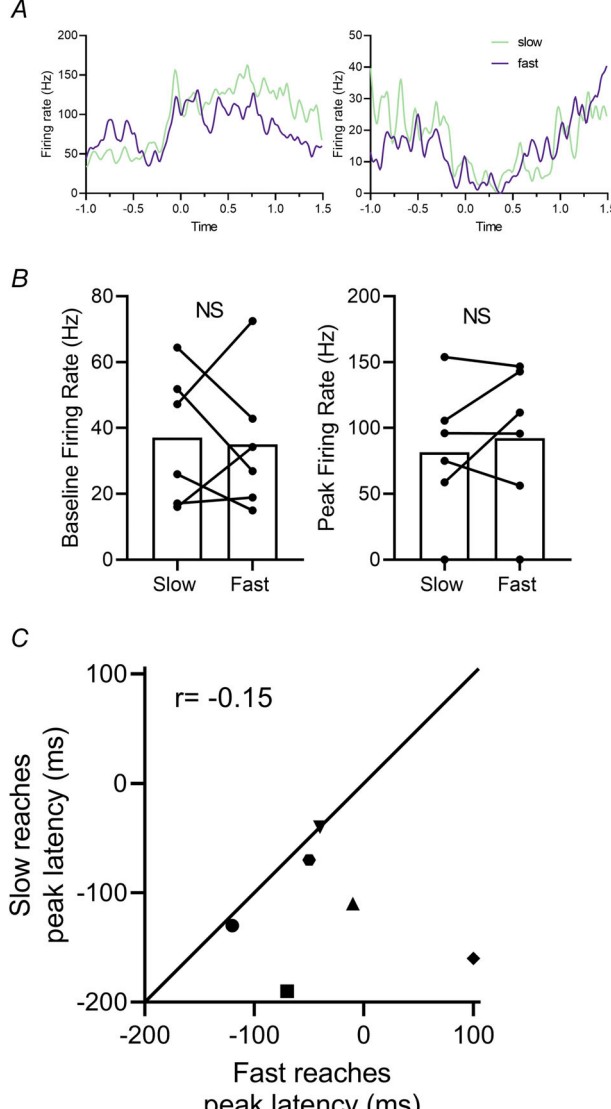

**Figure 7. Pattern of neural response is not related to duration of reach**

*A*, two example Purkinje cell simple spike units with PETHs constructed for fast (purple) *versus* slow reaches (green) obtained in the same recording session. PETHs aligned to paw lift (time zero). *B*, quantification of baseline firing rate and peak response rate during fast *versus* slow reaches (paired *t* test, *P* = 0.392, *n* = 6 units). *C*, scatterplot comparing average peak latency of unit response for fast and slow reaches within the same recording session (*r* = −0.15, *P* = 0.779). Each unit is represented by a different filled symbol; open filled symbol represents the peak latency for all reaches in the same recording session. [Colour figure can be viewed at wileyonlinelibrary.com]

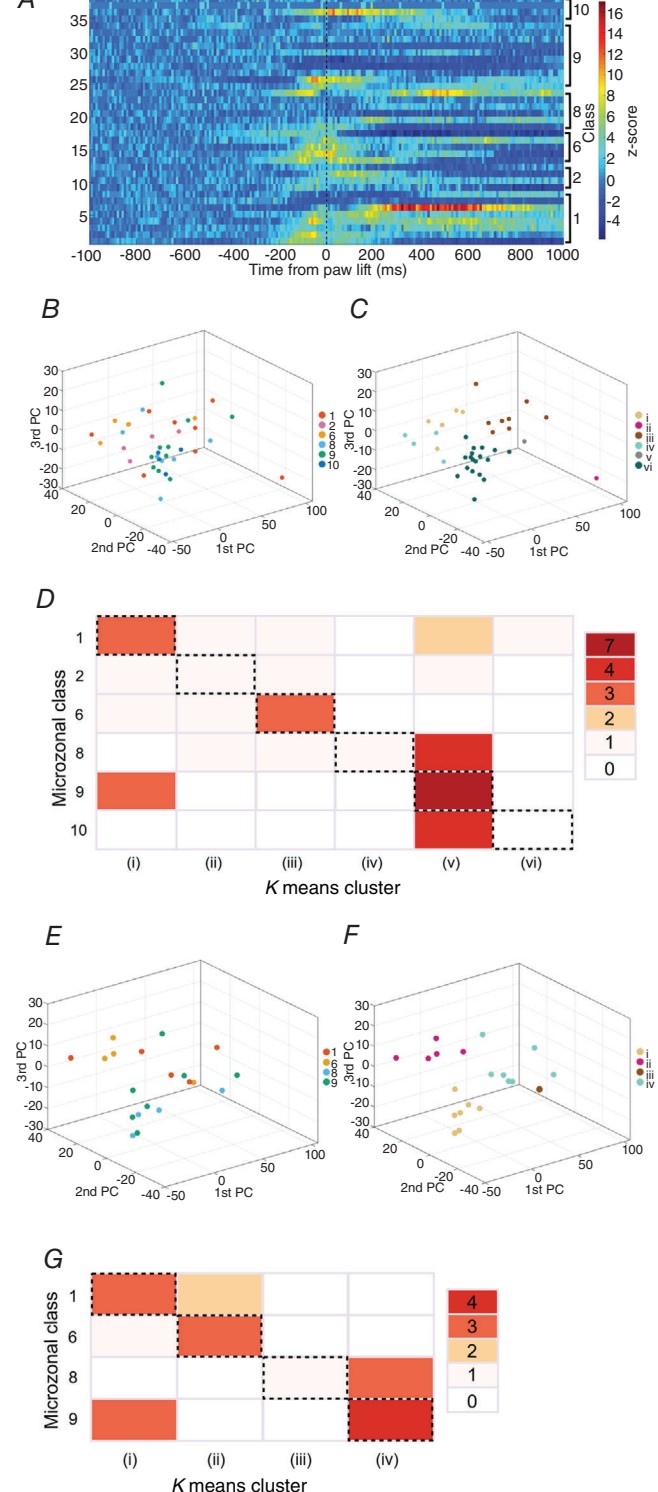

**Figure 8. Consistency of pattern of modulation of units within each class**

*A*, the activity patterns of units for which classes had four or more units (*n* = 38, left hand *y* axis) shown as a heat map and arranged in relation to receptive class (right hand *y* axis). Each row indicates the activity pattern of a single unit aligned to paw lift-off (time zero, vertical dashed line). Firing rates are colour coded according to

*z*-score values (colour scale to the right). *B*, projection in three dimensions of the first three principal components of the normalized *z*-scored spike firing rates for individual units ($n = 35$). Data points colour coded according to receptive field class (Class 1, 2, 6, 8, 9 and 10). *C*, *k*-means clustering of the first three principal components. The number of clusters ($n = 6$) was based on the number of receptive field classes with at least four available units to run the comparison. Data points colour coded according to *k*-means cluster (cluster i–vi). *D*, confusion matrix showing the relationship between *k*-means clustering of the first three principal components of unit activity shown in *A* compared to grouping units based on receptive field class. The *k*-means clustering is divided into six columns (i–vi) plotted against six receptive field classes of individual units. The grid squares on the diagonal (highlighted with dashed borders) show the number of units where the *k*-means clustering matched receptive field class. *E*, projection in three dimensions of the first three principal components of the normalized *z*-scored spike firing rates for Purkinje cells ($n = 20$). Data points colour coded according to receptive field class (Class 1, 2, 6, 8, 9 and 10). *F*, *k*-means clustering of the first three principal components. The number of clusters ($n = 4$) was based on the number of receptive field classes with at least four available units to run the comparison. Data points colour coded according to *k*-means cluster (cluster i–vi). *G*, confusion matrix showing the relationship between *k*-means clustering of the first three principal components of unit activity for Purkinje cell population for classes which had four or more units (classes 1, 6, 8 and 9). The *k*-means clustering is divided into four columns (i–iv) plotted against four classes of individual units. The grid squares on the diagonal show the number of units where the *k*-means clustering matched receptive field class. [Colour figure can be viewed at wileyonlinelibrary.com]

ARI = 96.3% while for the Purkinje cell population used in Fig. 8*C* ARI = 98.3%. The response profiles of the individual units as determined by PCA were therefore broadly related to receptive field class, and this was most evident for Purkinje cells.

### Receptive field classes and cortico-nuclear projections

Within the C3 zone Purkinje cells belonging to the same subzonal cortical region have cortico-nuclear projections that converge on common territories within nucleus interpositus anterior (Apps & Garwicz, 2000). We therefore used receptive field class as the basis to sum the simple spike activity of different subsets of Purkinje cells in order to estimate the pattern of modulation within the different cortico-nuclear pathways to nucleus interpositus anterior. Because of the low firing rates of Purkinje cell complex spikes which typically occur at about 2 Hz in the awake behaving animal (e.g. Armstrong & Rawson 1979; Horn et al., 2004) we were unable to collate sufficient numbers during task performance to quantify their activity. Instead, we focused on Purkinje cell simple spike activity as they carry the vast majority of information in the cortico-nuclear projection (Jorntell 2017) due to their wide dynamic firing range (Arenz et al., 2008; Garwicz et al., 1998; Prsa et al., 2009; van Kan et al., 1993). The baseline simple spike firing rate for a sample of Purkinje cells was 41.3 ± 21 Hz ($n = 24$). This is comparable to previous studies in cats (Armstrong & Rawson, 1979; Bengtsson et al., 2011; Cerminara et al., 2009; Edgley & Lidierth, 1988; Marple-Horvat & Criado, 1999; Yartsev et al., 2009).

As a first step, Fig. 9*A* illustrates the population activity for all available Purkinje cells ($n = 26$) recorded during task performance in relation to paw lift (hatched vertical line, Fig. 9*A*). The group PETH gives an estimate of the total pattern of simple spike modulation of Purkinje cells within the forelimb part of the C3 zone studied in the present experiments, and demonstrates that as a population, onset and peak of activity precedes paw lift, but activity remains elevated for ∼600 ms afterwards (including paw entry to the Perspex tube at ∼300 ms).

The patterns of simple spike activity of Purkinje cells within each available receptive field class were also combined to produce a single PETH per class (class 1, $n = 5$ Purkinje cells; class 2, $n = 2$ Purkinje cells; class 6, $n = 4$ Purkinje cells; class 8, $n = 4$ Purkinje cells; class 9, $n = 7$ Purkinje cells; class 10, $n = 2$ Purkinje cells; Fig. 9*B*). Onset and peak latencies were grouped into distal (digits, paw and wrist, classes 1, 2 and 6) and proximal (elbow, upper forelimb, shoulder, classes 8−10) location (Fig. 9*C*). There was a statistically significant difference in onset and peak latencies (proximal onset = 135.6 ±15.8 ms, $n = 9$; distal onset = −57 ± 160.6 ms, $n = 20$; proximal peak = 193.3 ± 121, $n = 9$; distal peak = 27.5 ± 136.6, $n = 20$; Mann–Whitney onset latency U = 147.5, $P = 0.005$, peak latency U = 143.5, $P = 0.01$).

The reaching component of the movement can be divided into two main phases: (i) paw lift, when there is initial flexion of the metacarpal phalangeal joints, followed by the elbow as the limb is raised from the surface (Martin et al., 1995) – this initial phase coincides with the increases in population simple spike activity of Purkinje cells belonging to receptive field classes 1, 6 and 9 (lower row of PETHs in Fig. 9*B*); and (ii) the reach phase, involving (iia) dorsiflexion of the paw and then (iib) protraction of the shoulder and scapula as the limb is elevated towards the target. This second phase coincides with increases in population simple spike activity of Purkinje cells belonging to class 2 (iia) and classes 8 and 10 (iib, left-hand column of PETHs in Fig. 9*B*). The results are therefore consistent with Purkinje cells belonging to different receptive field classes within the C3 zone collectively encoding different aspects of the reaching movement.

### Discussion

Our results demonstrate that the fine-grained organization of the C3 zone, originally established in anaesthetized and decerebrate preparations (Ekerot &

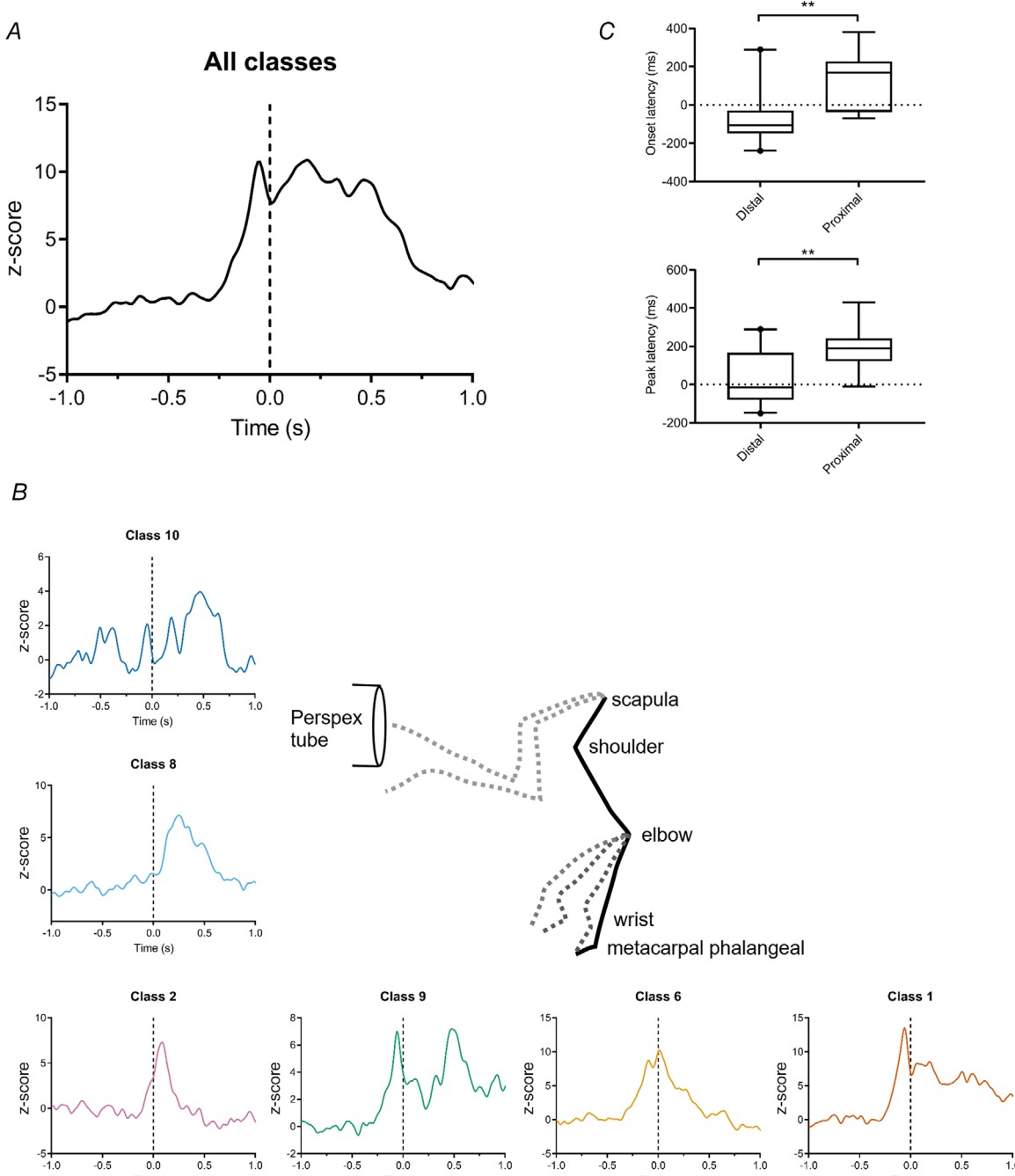

**Figure 9. Summed pattern of activity during reaching for Purkinje cells within the same receptive field class**

*A*, summed and *z*-scored PETH of Purkinje cell activity for all available receptive field classes aligned to paw lift (time zero, vertical dashed line, *n* = 26 Purkinje cells). *B*, stick figure showing movement trajectory of different segments of the forelimb during reaching to a target. Summed and *z*-scored PETH of Purkinje cell activity for each available receptive field class aligned to paw lift (time zero, vertical dashed line). In clockwise direction the PETHs are in order of onset latency (class 1, *n* = 5 Purkinje cells; class 2, *n* = 2 Purkinje cells; class 6, *n* = 4 Purkinje cells; class 8, *n* = 4 Purkinje cells; class 9, *n* = 7 Purkinje cells; class 10, *n* = 2 Purkinje cells). *C*, onset and peak latencies of Purkinje cell activity for each class in relation to paw lift. Data are presented as box and whisker plots, central horizontal bar is the median, and top and bottom horizontal lines of the box indicate upper and lower quartiles. Whiskers indicate 95% confidence intervals. Values on whisker limits are shown as individual data points. **\*\****P* < 0.01. [Colour figure can be viewed at wileyonlinelibrary.com]

Jorntell, 2001; Ekerot et al., 1991b; Garwicz et al., 1998), is also present in the awake animal. More specifically, our findings support the notion that the forelimb part of the C3 zone in the anterior lobe of the cerebellum has a detailed microzonal organization, which provides a structural and functional framework for cerebellar control of skilled limb movements.

The constraints of single unit recording in awake, non-head-fixed animals performing a skilled reaching task precluded a comprehensive analysis of the relationship between the spatial organization of the receptive field of units within subclasses (i.e. at the microzonal level of resolution) and their patterns of activity during reaching movements. Moreover, the small sample size of different cell types precluded separate analysis. Despite these limitations, our study nevertheless represents a first step towards a more detailed under-standing of information processing in the paravermal cerebellar cortex. Our results suggest that units belonging to different receptive field classes (i.e. groups of micro-zones with different sensory inputs) become active at different onset and peak latencies during the performance of a reaching task. By contrast, units belonging to the same class of receptive fields (i.e. groups of microzones with similar sensory inputs) displayed rather consistent activity patterns during reaching, at least in some receptive field classes (e.g. classes 6 and 9). It remains to be determined whether the lower degree of consistency in activity patterns among units belonging to some receptive field classes (e.g. classes 8 and 10) was due to variability between units within the same microzone, or whether, within the present sample, these classes contained a larger variety of individual microzones. Consistent with the latter possibility was the finding that units within classes 6 and 9 had less variation in RF location than those identified in classes 8 and 10.

### Simple spike and complex spike receptive fields of the same Purkinje cell

There is disagreement in the literature as to whether Purkinje cell simple spikes are activated by non-local groups of granule cells originating outside of a micro-zone (Ekerot & Jorntell, 2001) or from underlying, local granule cells (Bower & Woolston, 1983; Brown & Bower, 2001; Cohen & Yarom, 1998; Isope & Barbour, 2002). Whilst the limited sample size of Purkinje cells in which we were able to map both simple spike and complex spike receptive fields in the awake animal precludes a definitive result, nonetheless, our findings are in keeping with more recent studies which have demonstrated that Purkinje cells can be activated by both local and distant granule cell inputs (Cramer et al., 2013; Valera et al., 2016). Two-thirds of our sample of simple spikes and complex spikes recorded from the same Purkinje cells had

very similar receptive field characteristics, implying local granule cell input, but the remainder differed markedly, implying non-local inputs. Valera et al. (2016) found that plasticity-inducing protocols could activate previously quiescent granule cell synapses, altering the pattern of connections between granule cells and Purkinje cells. Also, large changes in C3 zone receptive field maps have been reported by electrically stimulating parallel fibres in decerebrate cats (Jorntell & Ekerot, 2002). Taken together these findings therefore raise the possibility that the receptive field characteristics of Purkinje cells can be shaped by behavioural experience. Future studies in awake animals will be needed to test this.

### To what extent is the contribution of the C3 zone to reaching behaviour organized in relation to microzones?

The results of the present study can be considered in relation to the hypothesis that within the C3 zone the concerted activity of microzones with different receptive fields combine to co-ordinate and control multijoint movements of the whole limb. First, receptive fields in the awake animal were found to be eccentric and focused around joints. This is in keeping with the original detailed mapping studies in anaesthetized and decerebrate animals (cf. Ekerot & Jorntell, 2001; Ekerot et al., 1991b; Garwicz et al., 1998) that suggested that the C3 zone receives integrated sensorimotor information related to the action of elemental movements (Apps & Garwicz, 2005; Ekerot et al., 1991b; Garwicz 2002; Garwicz et al., 2002). If such information was purely sensory then it might be expected that receptive fields would be more regular in shape, vary in size according to the density of cutaneous afferents and display a continuous representation of the skin of the limb, rather than – as in the case of the C3 zone – being divisible into *ca.* 40 microzones, thus approximately matching the number of muscles in the limb. On the other hand, the present findings suggest that the relationship between receptive field class and activity pattern is not always straightforward, but may display variability for some (sub)classes.

Second, the ARI was highest when correlation matrix analysis was restricted to Purkinje cells within the same class. This would imply that cerebellar cortical output is tuned in relation to receptive field identity, although a more complex relationship between receptive field and neuronal activity – and thereby between cerebellar cortical input and output – cannot be ruled out.

Third, microstimulation of nucleus interpositus has previously shown that individual microcomplexes within the C3 module control multisegmental muscle synergies (Ekerot et al., 1995), usually with a specific relationship between the receptive field of the microcomplex and the distal component of the movement evoked. For example,

stimulation of cerebellar nuclear sites with receptive fields located on the ventral paw and forearm were associated with dorsiflexion of the wrist; stimulation of sites with receptive fields on the medial forearm evoked elbow extension; and stimulation of sites with receptive fields located on the lateral paw and forearm produced flexion of the forearm; and stimulation of sites with receptive fields confined to the upper arm evoked shoulder flexion (Ekerot et al., 1995).

This functional activation can be mapped onto the present results. For example, class 6 and 9 Purkinje cells, which have receptive fields located on the lateral and radial paw and forearm respectively had population activity that was related to forearm extension; class 2 with receptive fields on the paw had population activity that was related to dorsiflexion of the wrist; and class 8 and 10 with receptive fields located on the upper arm, had population activity related to shoulder flexion. Therefore, the microzonal organization of the C3 zone appears to be matched by neuronal patterns of activity that reflect the control of different limb segments.

Fourth, multijoint control of synergistic movements by the cerebellum is supported by many studies (Becker & Person, 2019; Cooper et al., 2000; Fortier et al., 1989; Hoogland et al., 2015; Marple-Horvat & Stein, 1987; Thach et al., 1992) and multijoint movements have been shown to be particularly sensitive to cerebellar damage or experimental inactivation (Bastian et al., 1996; Becker & Person, 2019; Cooper et al., 2000; Hoogland et al., 2015; Low et al., 2018; Martin et al., 2000; Topka et al., 1998). In line with these previous studies we found, in one animal, that temporary inactivation of the cortical region under investigation produced deficits in the multijoint movement. While caution is clearly needed in drawing any firm conclusion from a single case, nonetheless this finding is consistent with the interpretation from previous studies that information processing within this cerebellar cortical region coordinates multijoint movements.

Fifth, microcomplexes have been suggested to coordinate movements either at the level of the cerebellar cortex through parallel fibres running across multiple microzones (Apps & Garwicz, 2005; Garwicz & Andersson, 1992), or by acting in concert on their target cerebellar nuclear cells (Ito, 1984). Previous studies have shown that groups of Purkinje cells firing synchronously can modulate their target cerebellar nuclear cells (Bengtsson et al., 2011; Blenkinsop & Lang, 2011; Person & Raman, 2011, 2012; Tang et al., 2019). Moreover, graded activation of Purkinje cell activity has been shown to result in a proportional inhibition of cerebellar nuclei in conjunction with changes in initiation, timing and pattern of movement (Heiney et al., 2014; Hoogland et al., 2015; Sarnaik & Raman, 2018). Given the high degree of convergence in the cortico-nuclear projection (Apps & Garwicz, 2005), and the number of receptive fields in

nucleus interpositus anterior is somewhat smaller than the number of microzones identified in the C3 zone (Garwicz & Ekerot, 1994), it is possible that microzones belonging to the same class, and therefore with similar receptive fields, converge on common groups of nuclear neurons. Our study found that individual Purkinje cells could vary in their pattern of task-related activity, but when we pooled activity of Purkinje cells belonging to the same receptive field class (i.e. potentially belonging to multiple microzones), we found systematic differences between classes in task-related activity. Purkinje cells with similar receptive fields may therefore act as an assembly on their target nuclear cells to control cerebellar output.

The overall picture that emerges is therefore one of a relatively specific relationship between patterns of movement-related activity and peripheral receptive fields. This implies that the representation of features of the whole movement by subgroups of cerebellar cortical neurons is organized in relation to sensory input, that is that neurons with similar receptive fields have similar patterns of activity and neurons with different receptive fields have different activity patterns, supporting functional organization in relation to microcomplexes. However, there are two important caveats: our study is limited to investigation of a single well-rehearsed behaviour – motor learning is likely to shape activity patterns; and if the C3 zone is organized in relation to different aspects of movement then it might be expected that activity of a given class of units would be selective to specific features of different tasks. Future studies will be needed to address these issues. Similarly, to establish a causal relationship between activity patterns and specific aspects of movements would require targeted intervention of Purkinje cells belonging to a specific receptive field class during task performance.

In summary, the current study advances our understanding of cerebellar contributions to behaviour by demonstrating that the C3 zone displays relatively specific patterns of neuronal activity in relation to its microzonal organization, consistent with operating within an integrated sensorimotor frame of reference, based on functional microcomplexes (Ito, 1984). To understand cerebellar information processing and its mode of operation as a coordinator we therefore suggest that a full description of receptive fields, topographical organization and neuronal activity in relation to well-defined behaviours is required.

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

## Additional information

### Data availability statement

The raw datasets generated and analysed during the current study are available from the following University of Bristol data repository, data.bris, at https://doi.org/10.5523/bris.2kgpk2eeewino2eka4kk8xa61r

### Competing interests

None.

### Author contributions

All authors contributed to the design and performance of the experiments as well as analysis of data and interpretation of findings. N.L.C. prepared the first draft of the manuscript and the figures. All authors reviewed the manuscript for intellectual content. The final version of manuscript for publication was explicitly approved by all authors.

### Funding

This work was supported by the Medical Research Council UK (G1100626) to N.L.C. and R.A. H.D. was supported by a BBSRC SWBio DTP Award (1503834).

### Acknowledgements

The authors are grateful for the support and assistance of Dr Jo Murrell (veterinary anaesthetist) and Rachel Bissett.

## Author's present address

Henry Darch: APC Microbiome Ireland, University College Cork, Cork, Ireland.

## Keywords

cerebellum, neural activity, Purkinje cell, reaching, receptive field

## Supporting information

Additional supporting information can be found online in the Supporting Information section at the end of the HTML view of the article. Supporting information files available:

**Statistical Summary Document**
**Peer Review History**
**Figure S1**

