## [Peer Review History · The Journal of Physiology]

Neuronal activity patterns in microcircuits of the cerebellar cortical C3 zone during reaching

Nadia L Cerminara, Martin Garwicz, Henry Darch, Conor Houghton, Dilwyn E Marple-Horvat, and Richard Apps
DOI: 10.1113/JP282928

Corresponding author(s): Nadia Cerminara (n.cerminara@bristol.ac.uk)

The following individual(s) involved in review of this submission have agreed to reveal their identity: Ken D O'Halloran (Referee #1)

Review Timeline:

Submission Date:	03-Feb-2022
Editorial Decision:	29-Mar-2022
Revision Received:	27-Jun-2022
Editorial Decision:	05-Aug-2022
Revision Received:	01-Sep-2022
Accepted:	07-Oct-2022

Senior Editor: Richard Carson

Reviewing Editor: Vatsala Thirumalai

Transaction Report:

Dear Dr Cerminara,

Re: JP-RP-2022-282928 "Neuronal activity patterns in microcircuits of the cerebellar cortical C3 zone during reaching" by Nadia L Cerminara, Martin Garwicz, Henry Darch, Conor Houghton, Dilwyn E Marple-Horvat, and Richard Apps

Thank you for submitting your manuscript to The Journal of Physiology. It has been assessed by a Reviewing Editor and by 3 expert Referees and I am pleased to tell you that it is considered to be acceptable for publication following satisfactory revision.

The reports are copied at the end of this email. Please address all of the points and incorporate all requested revisions, or explain in your Response to Referees why a change has not been made.

NEW POLICY: In order to improve the transparency of its peer review process The Journal of Physiology publishes online as supporting information the peer review history of all articles accepted for publication. Readers will have access to decision letters, including all Editors' comments and referee reports, for each version of the manuscript and any author responses to peer review comments. Referees can decide whether or not they wish to be named on the peer review history document.

Authors are asked to use The Journal's premium BioRender (<https://biorender.com/>) account to create/redraw their Abstract Figures. Information on how to access The Journal's premium BioRender account is here:

<https://physoc.onlinelibrary.wiley.com/journal/14697793/biorender-access> and authors are expected to use this service. This will enable Authors to download high-resolution versions of their figures. The link provided should only be used for the purposes of this submission. Authors will be charged for figures created on this premium BioRender account if they are not related to this manuscript submission.

I hope you will find the comments helpful and have no difficulty returning your revisions within 4 weeks.

Your revised manuscript should be submitted online using the links in Author Tasks Link Not Available.

Any image files uploaded with the previous version are retained on the system. Please ensure you replace or remove all files that have been revised.

REVISION CHECKLIST:

- Article file, including any tables and figure legends, must be in an editable format (eg Word)
- Abstract figure file (see above)
- Statistical Summary Document
- Upload each figure as a separate high quality file
- Upload a full Response to Referees, including a response to any Senior and Reviewing Editor Comments;
- Upload a copy of the manuscript with the changes highlighted.

- A potential 'Cover Art' file for consideration as the Issue's cover image;
- Appropriate Supporting Information (Video, audio or data set https://jp.msubmit.net/cgi-bin/main.plex?form_type=display_requirements#supp).

To create your 'Response to Referees' copy all the reports, including any comments from the Senior and Reviewing Editors, into a Word, or similar, file and respond to each point in colour or CAPITALS and upload this when you submit your revision.

I look forward to receiving your revised submission.

If you have any queries please reply to this email and staff will be happy to assist.

Yours sincerely,

Richard Carson
Senior Editor
The Journal of Physiology

REQUIRED ITEMS:

- Author photo and profile. First (or joint first) authors are asked to provide a short biography (no more than 100 words for one author or 150 words in total for joint first authors) and a portrait photograph. These should be uploaded and clearly labelled with the revised version of the manuscript. See Information for Authors for further details.
- Please upload separate high-quality figure files via the submission form.
- Please ensure that any tables are in Word format and are, wherever possible, embedded in the article file itself.
- Please ensure that the Article File you upload is a Word file.
- A Statistical Summary Document, summarising the statistics presented in the manuscript, is required upon revision. It must be on the Journal's template, which can be downloaded from the link in the Statistical Summary Document section here: https://jp.msubmit.net/cgi-bin/main.plex?form_type=display_requirements#statistics
- Papers must comply with the Statistics Policy https://jp.msubmit.net/cgi-bin/main.plex?form_type=display_requirements#statistics

In summary:

- If $n \leq 30$, all data points must be plotted in the figure in a way that reveals their range and distribution. A bar graph with data points overlaid, a box and whisker plot or a violin plot (preferably with data points included) are acceptable formats.
- If $n > 30$, then the entire raw dataset must be made available either as supporting information, or hosted on a not-for-profit repository e.g. FigShare, with access details provided in the manuscript.
- 'n' clearly defined (e.g. x cells from y slices in z animals) in the Methods. Authors should be mindful of pseudoreplication.
- All relevant 'n' values must be clearly stated in the main text, figures and tables, and the Statistical Summary Document (required upon revision).
- The most appropriate summary statistic (e.g. mean or median and standard deviation) must be used. Standard Error of the Mean (SEM) alone is not permitted.
- Exact p values must be stated. Authors must not use 'greater than' or 'less than'. Exact p values must be stated to three significant figures even when 'no statistical significance' is claimed.
- Statistics Summary Document completed appropriately upon revision.
- A Data Availability Statement is required for all papers reporting original data. This must be in the Additional Information section of the manuscript itself. It must have the paragraph heading "Data Availability Statement". All data supporting the results in the paper must be either: in the paper itself; uploaded as Supporting Information for Online Publication; or archived in an appropriate public repository. The statement needs to describe the availability or the absence of shared data. Authors must include in their Statement: a link to the repository they have used, or a statement that it is available as Supporting Information; reference the data in the appropriate section(s) of their manuscript; and cite the data they have shared in the References section. Whenever possible the scripts and other artefacts used to generate the analyses presented in the paper should also be publicly archived. If sharing data compromises ethical standards or legal requirements then authors are not expected to share it, but must note this in their Statement. For more information, see our Statistics Policy.
- Please include an Abstract Figure. The Abstract Figure is a piece of artwork designed to give readers an immediate understanding of the research and should summarise the main conclusions. If possible, the image should be easily 'readable' from left to right or top to bottom. It should show the physiological relevance of the manuscript so readers can assess the importance and content of its findings. Abstract Figures should not merely recapitulate other figures in the manuscript. Please try to keep the diagram as simple as possible and without superfluous information that may distract from the main conclusion(s). Abstract Figures must be provided by authors no later than the revised manuscript stage and should be uploaded as a separate file during online submission labelled as File Type 'Abstract Figure'. Please ensure that you include the figure legend in the main article file. All Abstract Figures should be created using BioRender. Authors should use

The Journal's premium BioRender account to export high-resolution images. Details on how to use and access the premium account are included as part of this email.

EDITOR COMMENTS

Reviewing Editor:

This manuscript by Cerminara et al. describe activity patterns of Purkinje neurons during a forelimb reach task in cats and relate these activity changes to the receptive field of these same neurons. These data will be important for the field. Nevertheless, there are a few major weaknesses which have been highlighted by both the reviewers and the authors need to address these satisfactorily before the manuscript can be accepted. Please refer to the detailed reviews below and prepare a point-by-point response while submitting your revision.

REFEREE COMMENTS

Referee #1:

Thank you for submitting your manuscript to The Journal of Physiology. Some additional details pertaining to animal welfare are required.

Line 137: Give dose and route of administration of propofol. The stock concentration is given, but the authors need to provide the dose range administered to animals.

Line 138: Include details on the carrier gas for isoflurane anaesthesia.

Line 199: From earlier comments it appears that no head restraint was required during procedures. Please confirm that this also applied to this section, when microelectrodes were advanced into the brain. The authors have described procedures well and have given the required assurances that animal welfare standards were upheld in the course of training, surgery, post-surgical care and during recording sessions. It would be helpful to include an additional statement on animal care/welfare during the insertion of microelectrodes and the 22 gauge needle.

General: How many recording sessions were performed in each cat?

Referee #2:

This paper broadly addresses the question of how cerebellar neural activity contributes to the control of movement. A fundamental problem in addressing this question is to know the mapping between neurons within the cerebellum and functional activity in muscles, and this has been recognised for a very long time. The paper addresses the problem by relying on a long history of careful analysis of how the cerebellar cortex is organised, based on anatomy and electrophysiology, much of which has been done by the groups in Bristol and Lund. This work, as well as work by others, has shown that the cerebellar cortex can be subdivided into zones based on input and output, so that by identifying the zone we can know what the output of that part of the cerebellum is likely to be doing.

The strength of the paper is that it has been shown some time ago that the organisation within the C3 zone is not continuous but seems to have a discontinuous organisation where microzones and is have sensory inputs that mapped them into discrete groups with specific receptive fields, and much evidence points to these being unitary elements of fallen movements.

It is very noticeable that, despite the knowledge that zones and microzones exist being available for many decades, many studies of cerebellar function do not pay enough attention to the identification of neurons, and therefore do not directly address the question of how cerebellar activity relates to movement.

In the studies reported in this paper recordings made from neurons areas of cerebellar cortex which are functionally classified by micro zone during a forelimb reaching task - allowing the question of how the neural activity links to the task to be addressed directly.

The paper shows that there is a tendency for general similarity of timing and modulation patterns of Purkinje cells within each of the receptive field groupings, and provides an estimate of the Purkinje cell modulation provided to the nucleus interpositus anterior to which the micros own is from which the recordings made are focused.

The paper is carefully and precisely written.

Specific Points

1. I found the reference to the data showing that this part of the cerebellum being organised within a "movement frame of reference" somewhat confusing. What would the alternative be, based on the large amount of data already available on this area?

2. The reported data include Purkinje cells, with both climbing fibre and simple spike activity, units identified as mossy fibre terminals and also putative cortical interneurons including prospective Golgi cells. I think everyone knows that these experiments are extremely technically demanding, and it's natural therefore to want to include as much data as possible, however there are relatively small prospective mossy fibre terminals and prospective interneurons in the dataset, yet these are added in and not differentiated from Purkinje cell data in the results. I would question whether this is really helpful and whether it might be better simply to focus on Purkinje cell recordings. A number of issues arise:

i) The paper states that the peripheral receptive fields of these putative cortical interneurons also fell into the same categories as the Purkinje cell receptive fields. This is interesting because a number of studies have found much broader receptive fields of Golgi cells, including in awake animals: e.g. Dash et al *Journal of Neuroscience* 2009, 29 (1) 250-262: who found that mossy fibre terminals highly heterogeneous but were sharply tuned to metrics of eye movements whereas Golgi cells in the same area had discharges that were not strongly modulated or correlated in relation to the metrics of movements. The number of recordings here from mossy fibre terminals and prospective Golgi cells is small, but the suggestion that they simply follow the same fractionated receptive fields as Purkinje cells is strongly at odds with other observations on Golgi cells.

ii) Following from this, the section of the results on task performance and patterns of activity reports firing rates for groups of 'units' with particular receptive fields: for example in figures 5 and 6. Again one might expect these to be biased since the groupings include relatively small numbers of units, and some will include mossy fibre terminals or prospective Golgi cells, which might be expected to have very different firing rates and modulations (again there are a number of studies of Golgi cells showing that they tend to fire at much lower rates, and have much less dramatic rate modulation than Purkinje cells do). If mossy fibre terminals and prospective Golgi cells are included in these receptive field groups, then evidence needs to be presented that including diverse types of unit measures of rate is justifiable.

5. The section on comparison of firing of units with similar receptive fields could be much clearer: for example figure 8A is hard to read. I don't see why brown and pale blue were chosen for the Z score range rather than some stronger contrasting colours, and why no Z score is grey rather than white: the overall appearance is a little blurry. I also found figure 8 hard to read: isn't an alternative interpretation that my present classes 8,9 and 10 are more similar in activity (cluster v) than they are different?

6. In discussion line 856 on the paper states "In the present study the reaching movement involved multi-joint coordination and in one animal temporary inactivation of the cortical region under investigation produced deficits in the whole movement,". Its not clear to me what the 'whole movement' is or whether a single observation in a single animal is really appropriate.

Other Points

7. Figure 1: labels on the parts of the figure are quite sparse.

8. In D I was initially disoriented, since in many studies cerebellar sections are presented with rostral on the left-hand side and caudal on the right - if that were the case the area labelled lobule 5 would be in the posterior lobe. Markers identifying anterior posterior and some of the other lobules are needed. The small inset fluorescent labelled section is going to be extremely small in a final version of the figure, and I cannot see very clearly how it relates to the line drawing. This needs to be improved.

9. In various parts of this figure the scale bars are of different thickness and the lack of labels on the actual images means the reader has to keep jumping between the figure legend and the image just to check the scales: some consistency across the figures and putting the numbers on the images would really help with this.

10. In figure 7 comparison is made between firing patterns (of units - it would be much more valuable to know that these were Purkinje cells) during reaching movements of long and slow duration. Given that these are individual traces is a great deal of variation in firing rate: since we are really interested in the pattern of activity over the duration of the reach, would it not be better to smooth the firing rate? For example using a moving average or convolution? Also in the final version of the figure the current relatively noisy traces are going to make it hard to differentiate the grey from the black - wouldn't colour be more appropriate here?

Referee #3:

The manuscript by Cerminara et al. investigated the receptive field of the cerebellar Purkinje cells in the forelimb part of the C3 zone in awake cats and analyzed their firing patterns during the forelimb reaching task. I agree that it is important to know RF maps in the awake state and their relationship to the spiking activity during behavior, and the results will definitely be of interest to researchers working in the cerebellar field. However, unfortunately, the topic and the findings are very limited to the specific part of the cerebellum, and most of the data are qualitative. Therefore, they do not advance the field sufficiently to justify publication in the *Journal of Physiology*. I suggest that the manuscript would be better suited for a

specialized journal such as The Cerebellum.

I suggest the authors consider the following points before submission to another journal.

Receptive field mapping: The authors performed RF mapping by the classical method that they manually deliver stimuli by brushing and tapping the skin and so on. This established method has been used for anesthetized or decerebrated cats, in which volitional movements were lost. In awake cats, however, they may respond to the stimulus, and such behavioral responses should affect the results. The authors should describe how they controlled stimulus and behavior during RF mapping experiments in the awake state. Furthermore, all the RF maps have been created based on the very subjective measure, i. e., cells' "strongest responses" that must depend on the strength of the stimuli, and it may be difficult to control by manual stimulation. The information for the classification of the cell type is also very limited. The authors only described the criteria in the Materials and Methods section and did not show any data.

Reaching task: There is a little description of how well the cats performed the task. In particular, it is essential to know how typical the movement trajectories are, given that the authors compared the shape of PETH between different unit types in different cats. For example, in Figure 6, example PETHs are shown with their RF locations, but for the information about cats' behavior, only the time from paw lift is available.

Cluster analysis: The authors used principal component analysis followed by k-means clustering to classify the pattern of unit activity and showed that each k-means cluster corresponded to a specific receptive field class. However, the number of the k-means clusters can be determined arbitrarily, and the authors did not show the results of PCA and clustering so that no one can judge whether the results shown in Figure 8 and Table 1 are correct. In particular, Table 1 does not make sense. The authors should present not only the results of classification but the raw data or PCA/clustering results to show the rationale for the classification.

END OF COMMENTS

Confidential Review

03-Feb-2022

We thank the referees and editor for their insights and constructive comments. We have addressed all the points raised in the point by point reply below.

REFeree COMMENTS

Referee #1

- 1.1. Line 137: Give dose and route of administration of propofol. The stock concentration is given, but the authors need to provide the dose range administered to animals.
The range of the rate of administration (0.1-0.3mg/kg/min) has been included in the Methods (p6, Line 143).
- 1.2. Line 138: Include details on the carrier gas for isoflurane anaesthesia.
The carrier gas (oxygen) has now been included in the Methods (p6, Line 144).
- 1.3. Line 199: From earlier comments it appears that no head restraint was required during procedures. Please confirm that this also applied to this section, when microelectrodes were advanced into the brain. The authors have described procedures well and have given the required assurances that animal welfare standards were upheld in the course of training, surgery, post-surgical care and during recording sessions. It would be helpful to include an additional statement on animal care/welfare during the insertion of microelectrodes and the 22 gauge needle.
The animal's head was lightly held by an experimenter whilst another experimenter placed and secured the microdrive onto the chamber. This procedure lasted ~5s and was very well tolerated by the animals, as part of their habituation and training was to have their heads patted, stroked and lightly held. The animals did not show any signs of discomfort or distress when the microelectrode or needle were being advanced. We have added this information in the Methods (p7, Line 190, p8 Line 212 and p9 Lines 248 and 252).
- 1.4. General: How many recording sessions were performed in each cat?
On average, recording sessions occurred over a period of 8 weeks (range 4-12 weeks), 6 days per week. This information has been added to the Methods (p7, Line 189).

Referee #2

- 2.1. I found the reference to the data showing that this part of the cerebellum being organised within a "movement frame of reference" somewhat confusing. What would the alternative be, based on the large amount of data already available on this area?
We have amended the Introduction (p3, line 62; p4, line 95, p5, line 111) to try and make clear that, while it is widely recognised that the cerebellum is a sensorimotor structure, how that translates to functional organization at a microcircuit level during behaviour remains unclear. Previous work on cerebellar cortical neuronal activity during voluntary behaviour has not taken into consideration the microzonal topography. Our study shows there is a relationship between peripheral receptive field and the overall activity pattern of any given neuron within the forelimb-receiving part of the C3 zone, i.e. that neurons with similar receptive fields have similar patterns of activity and neurons with different receptive field have different activity patterns. The present findings therefore provide strong evidence for the validity of the microzonal organization of this part of the cerebellum. We agree that movement frame of reference does not capture this point so we have amended to 'microzonal frame of reference' This point has also been added to the Discussion (p33, line 789, p35, line 849). The use of "microzonal" or "microcomplex" depends on which studies are being referred to. However, for the purposes of the present study, these terms are interchangeable.
- 2.2. The paper states that the peripheral receptive fields of these putative cortical interneurons also fell into the same categories as the Purkinje cell receptive fields. This is interesting because a

number of studies have found much broader receptive fields of Golgi cells, including in awake animals: e.g. Dash et al *Journal of Neuroscience* 2009, 29 (1) 250-262: who found that mossy fibre terminals highly heterogeneous but were sharply tuned to metrics of eye movements whereas Golgi cells in the same area had discharges that were not strongly modulated or correlated in relation to the metrics of movements. The number of recordings here from mossy fibre terminals and prospective Golgi cells is small, but the suggestion that they simply follow the same fractionated receptive fields as Purkinje cells is strongly at odds with other observations on Golgi cells.

The reviewer asks about the difference in the receptive field properties of the putative Golgi cells in comparison to previous studies. Our Golgi cells were identified based on their long spike duration, low and regular firing rate and large tuning distance in line with many other studies (e.g. Barmack and Yakhnitsa, 2008; Holtzman et al. 2006; Simpson et al. 2005 and Edgley and Lidieth, 1987). Our study demonstrated that our putative Golgi cells had small receptive fields, whereas other have shown receptive fields to much broader (Holtzman et al., 2006; Xu and Edgley 2010; Vos et al., 2008); in some cases from all four limbs and face in rats. Whilst we agree that there is a difference between our studies and those cited, the studies are not directly comparable: Holtzman et al., and Xu and Edgley used electrical stimulation and the study by Vos et al examined Golgi responses in the rat to whisker stimulation. The study by Prsa, Dash and others (Prsa et al., 2009) raised by the reviewer examined Golgi cell activity in relation to saccadic eye movement which again is not directly comparable to our study. Moreover, the study by Ekerot and Jorntell, 2001 (see their Fig. 4B) demonstrated that the receptive fields of putative Golgi cells had restrictive receptive fields similar to what was found in our study, therefore we do not believe our putative Golgi cells are misclassified and are in keeping with what is known about Golgi cells responses and microzones. However, since we cannot be definitive about the cell identify of interneurons, we use the term 'putative' (p8 lines 221 and 224).

- 2.3. Following from this, the section of the results on task performance and patterns of activity reports firing rates for groups of 'units' with particular receptive fields: for example in figures 5 and 6. Again one might expect these to be biased since the groupings include relatively small numbers of units, and some will include mossy fibre terminals or prospective Golgi cells, which might be expected to have very different firing rates and modulations (again there are a number of studies of Golgi cells showing that they tend to fire at much lower rates, and have much less dramatic rate modulation than Purkinje cells do). If mossy fibre terminals and prospective Golgi cells are included in these receptive field groups, then evidence needs to be presented that including diverse types of unit measures of rate is justifiable.

The original classification system of the C3 microzones based on the climbing fibre system was carried out by Ekerot et al., (1991) in the anaesthetised cat. Several studies in different species have demonstrated that mossy fibres can also have a sagittal termination pattern respecting the zonal organisation of climbing fibre inputs (e.g. Hazlett et al. 1971; Ekerot & Larson, 1980; Gerrits et al. 1985; Heckroth & Eisenman, 1988; Serapide et al. 1994; Pijpers et al., 2006). A direct comparison with climbing fibre microzones was carried out by Garwicz et al (1998) in the anaesthetised cat. Garwicz et al., (1998) found that the relationship between mossy fibre input and the climbing fibre input to the same cortical regions had a high degree of similarity in spatial organization of receptive fields and sequences of termination within the zone for the two types of afferents. Thus an anatomical and physiological framework exists to group PCs and mossy fibres with similar receptive fields. In terms of molecular layer interneurons, a study by Ekerot and Jorntell (2001) again in the anaesthetised cat has shown that they also have similar receptive field classes to the climbing fibre system. For this reason, we have used the peripheral location and original class system as our framework for classifying our units into classes.

In terms of activity profiles, all previous studies have focused on the similarity and differences of receptive field spatial organization. To our knowledge, our study represents the first attempt to study the activity patterns of different unit types within a receptive field class in the awake behaving animal. We have grouped the unit types of each class regardless of the activity profiles, so long as they belonged to the same class, as during the task, stimulation of the receptive field will activate all units within that class, regardless of whether they have an increase or decrease in response profile, and given the anatomical and physiological framework outlined above they are likely to act in a concerted way during behaviour. We have indicated in the figure legends and on the figures what type of unit (i.e. PC simple spike, MF etc). The PCA and Adjusted Rand Index (ARI) analysis uses z-scored data which normalises for differences in firing rate. We carried out this analysis on all unit types as well as just the Purkinje cells. The ARI was 96.8% and 98.3% for all units and Purkinje cells only, respectively, which indicates that the response profiles of the units are broadly related to the receptive field class. The ARI is higher for the Purkinje cells only as this is most likely due to having more neurons of this type.

Our reason for pooling activity data from all types of neurons and neuronal elements within microzones was the small sample sizes. If we had sufficient n-numbers of each of mossy fibres, PCs and Golgi cells, we would have analysed these separately. We admit these limitations of the data set. However, despite these limitations, our study represents a first step towards a more detailed understanding of the information processing in this part of the cerebellum, taking into consideration its functional connectivity. Notably, in particular for Golgi cells, the contribution to the pooled patterns of activity is minor, due to low firing rates and small number of cells. This point has been included in Discussion (p32, line 759).

- 2.4 The section on comparison of firing of units with similar receptive fields could be much clearer: for example figure 8A is hard to read. I don't see why brown and pale blue were chosen for the Z score range rather than some stronger contrasting colours, and why no Z score is grey rather than white: the overall appearance is a little blurry. I also found figure 8 the hard to read: isn't an alternative interpretation that my present classes 8,9 and 10 are more similar in activity (cluster v) than they are different?

We have changed the colour scheme for panel A Figure 8. In relation to the comment raised about interpretation of classes 8, 9 and 10 and the confusion matrix, we agree that there is some 'confusion' between 8, 9 and 10 and between 1 and 9; this may be explained by these classes of neuron being involved in similar parts of the movement. This point has now been included in the manuscript (p26, line 611).

- 2.5. In discussion line 856 on the paper states "In the present study the reaching movement involved multi-joint coordination and in one animal temporary inactivation of the cortical region under investigation produced deficits in the whole movement,". Its not clear to me what the 'whole movement' is or whether a single observation in a single animal is really appropriate. We used the term whole movement to mean multi-joint; this has now been changed in the main text (p33, line 792). The inclusion of the single animal observation follows on from a long line of evidence from other studies. We have altered the sentence accordingly to make it clearer that we are not making this claim based on just our one example, but the finding is consistent with a comprehensive set of other previous evidence (p34, Line 827).

- 2.6. Figure 1: labels on the parts of the figure are quite sparse. In D I was initially disoriented, since in many studies cerebellar sections are presented with rostral on the left-hand side and caudal on the right - if that were the case the area labelled lobule 5 would be in the posterior lobe. Markers identifying anterior posterior and some of the other lobules are needed. The small inset fluorescent labelled section is going to be extremely small in a final version of the figure, and

I cannot see very clearly how it relates to the line drawing. This needs to be improved. In various parts of this figure the scale bars are of different thickness and the lack of labels on the actual images means the reader has to keep jumping between the figure legend and the image just to check the scales: some consistency across the figures and putting the numbers on the images would really help with this.

We thank the reviewer for spotting the inconsistency and have now made the scale bars all the same point size. We have also added the scale bar values to the figure to avoid having to look at the figure legend. We have also added extra labelling to panel D and flipped the panel around so that anterior/rostral is now towards the left and posterior/caudal is towards the right. We have also added an arrow to make it clearer that the photomicrograph inset relates to the camera lucida drawing and have made the photo larger.

- 2.7. In figure 7 comparison is made between firing patterns (of units - it would be much more valuable to know that these were Purkinje cells) during reaching movements of long and slow duration. Given that these are individual traces is a great deal of variation in firing rate: since we are really interested in the pattern of activity over the duration of the reach, would it not be better to smooth the firing rate? For example using a moving average or convolution? Also in the final version of the figure the current relatively noisy traces are going to make it hard to differentiate the grey from the black - wouldn't colour be more appropriate here?

We have now made it explicit that this analysis is related to Purkinje cells in the text and Figure legend (p24, Line 573 and Figure Legend 7). We have also smoothed the PETH using a gaussian kernel of 10ms and used colours for the traces. The findings from this analysis remain unchanged.

Referee #3

- 3.1. Receptive field mapping: The authors performed RF mapping by the classical method that they manually deliver stimuli by brushing and tapping the skin and so on. This established method has been used for anesthetized or decerebrated cats, in which volitional movements were lost. In awake cats, however, they may respond to the stimulus, and such behavioral responses should affect the results. The authors should describe how they controlled stimulus and behavior during RF mapping experiments in the awake state. Furthermore, all the RF maps have been created based on the very subjective measure, i. e., cells' "strongest responses" that must depend on the strength of the stimuli, and it may be difficult to control by manual stimulation. The information for the classification of the cell type is also very limited. The authors only described the criteria in the Materials and Methods section and did not show any data.

Eight of the nine animals were bred in house i.e. were in our animal house from birth. The remaining animal was purchased and imported into our animal facility as a kitten. Whilst we were waiting for the animals to reach young adulthood and prior to training in the reach task, time was spent each day ensuring that the animals were used to being patted, tapped and stroked in a way that mimicked non-noxious receptive field mapping. Therefore, our animals were very well habituated to somatosensory testing from kittens through to adulthood, and as such, the animals remained passive during receptive field mapping and did not show any behavioural responses i.e. they would sit quietly during testing. This information has now been added to the Methods (p5, Line 126).

Additionally, as noted in the Methods (Page 9, Line 237) receptive field mapping was carried midway through the task to check if there were any changes to the receptive field and none were noted, therefore we are reasonably confident that our somatosensory testing was delivered in a consistent way. The experiments were carried out with at least two experimenters (and in many cases 3) who would reach consensus about the receptive field. Any units which we could not

clearly identify in terms of the receptive field location remained unclassified. One of the authors (Garwicz) was one of the original authors who carried out the initial receptive field mapping of the C3 microzones in the anaesthetised cat and therefore has many years of expertise determining the receptive fields. Therefore, our careful and conservative approach when testing and classifying the units means that we are confident that receptive field map testing was unaffected by behavioural responses or alterations in the strength of the stimuli delivered (p9, line 242).

In response to the reviewer, we have also added examples of putative Golgi cells and mossy fibres to Figure 1.

- 3.2. Reaching task: There is a little description of how well the cats performed the task. In particular, it is essential to know how typical the movement trajectories are, given that the authors compared the shape of PETH between different unit types in different cats. For example, in Figure 6, example PETHs are shown with their RF locations, but for the information about cats' behavior, only the time from paw lift is available.

The reaching task is based on the study by Alstermark et al., (1986, 1993) and Schepens and Drew, 2003). The animals were able to learn the task and were successfully able to reach into the tube to retrieve the food reward without any difficulty, as they associated the task with extra food rewards and attention from the experimenters. As stated in results (p23, line 557) these previous studies in cats have shown that the reaching task used is highly stereotypical, with little variation between trials in terms of movement trajectories and velocity with cats adopting a very similar kinematic reaching strategy. We focussed our analysis to the paw lift and reaching to the tube because other events later in the task such as tube exit, paw down etc were considerably more variable in timing. The only time we observed a change in reaching was towards the end of the recording session when the cats were becoming satiated. When this occurred, the animals tended to slow down their movements and we generally stopped the recording session. This point has been added to Results (p23, line 558). To investigate whether differences in movement speed were a factor we compared the duration of the reach (taken from time of paw lift to entry into the tube, see Methods) for each receptive field class. Our analysis found no evidence to suggest that the differences in pattern of neuronal activity observed between receptive field classes was related to differences in reach performance (p24 line 564).

- 3.3. Cluster analysis: The authors used principal component analysis followed by k-means clustering to classify the pattern of unit activity and showed that each k-means cluster corresponded to a specific receptive field class. However, the number of the k-means clusters can be determined arbitrarily, and the authors did not show the results of PCA and clustering so that no one can judge whether the results shown in Figure 8 and Table 1 are correct. In particular, Table 1 does not make sense. The authors should present not only the results of classification but the raw data or PCA/clustering results to show the rationale for the classification.

We have now included the PCA and k-means in Figure 8 and have added the associated text in the figure legend. In k-means clustering the number of clusters is, indeed, arbitrary; we were interested in comparing the clustering determined by receptive field and the firing pattern of the units during behaviour. Thus, we wanted to ask what is the clustering implied by using the same number of clusters for the spike trains as we believe are present because of the grouping of the units by receptive field. We are not claiming to have determined the number of clusters based on spiking, we are applying the cluster number from the receptive fields to spiking. This point has been made clearer (p26, line 602).

Dear Dr Cerminara,

Re: JP-RP-2022-282928R1 "Neuronal activity patterns in microcircuits of the cerebellar cortical C3 zone during reaching" by Nadia L Cerminara, Martin Garwicz, Henry Darch, Conor Houghton, Dilwyn E Marple-Horvat, and Richard Apps

Thank you for submitting your manuscript to The Journal of Physiology. It has been assessed by a Reviewing Editor and by 2 expert Referees and I am pleased to tell you that it is considered to be acceptable for publication following satisfactory revision.

The reports are copied at the end of this email. Please address all of the points and incorporate all requested revisions, or explain in your Response to Referees why a change has not been made.

NEW POLICY: In order to improve the transparency of its peer review process The Journal of Physiology publishes online as supporting information the peer review history of all articles accepted for publication. Readers will have access to decision letters, including all Editors' comments and referee reports, for each version of the manuscript and any author responses to peer review comments. Referees can decide whether or not they wish to be named on the peer review history document.

Authors are asked to use The Journal's premium BioRender (<https://biorender.com/>) account to create/redraw their Abstract Figures. Information on how to access The Journal's premium BioRender account is here:

<https://physoc.onlinelibrary.wiley.com/journal/14697793/biorender-access> and authors are expected to use this service. This will enable Authors to download high-resolution versions of their figures. The link provided should only be used for the purposes of this submission. Authors will be charged for figures created on this premium BioRender account if they are not related to this manuscript submission.

I hope you will find the comments helpful and have no difficulty returning your revisions within 4 weeks.

Your revised manuscript should be submitted online using the links in Author Tasks: Link Not Available.

Any image files uploaded with the previous version are retained on the system. Please ensure you replace or remove all files that have been revised.

REVISION CHECKLIST:

- Article file, including any tables and figure legends, must be in an editable format (eg Word)
- Abstract figure file (see above)
- Statistical Summary Document
- Upload each figure as a separate high quality file
- Upload a full Response to Referees, including a response to any Senior and Reviewing Editor Comments;
- Upload a copy of the manuscript with the changes highlighted.

- A potential 'Cover Art' file for consideration as the Issue's cover image;
- Appropriate Supporting Information (Video, audio or data set https://jp.msubmit.net/cgi-bin/main.plex?form_type=display_requirements#supp).

To create your 'Response to Referees' copy all the reports, including any comments from the Senior and Reviewing Editors, into a Word, or similar, file and respond to each point in colour or CAPITALS and upload this when you submit your revision.

I look forward to receiving your revised submission.

If you have any queries please reply to this email and staff will be happy to assist.

Yours sincerely,

Richard Carson
Senior Editor
The Journal of Physiology

REQUIRED ITEMS:

- Author photo and profile. First (or joint first) authors are asked to provide a short biography (no more than 100 words for one author or 150 words in total for joint first authors) and a portrait photograph. These should be uploaded and clearly labelled with the revised version of the manuscript. See Information for Authors for further details.
- The Journal of Physiology funds authors of provisionally accepted papers to use the premium BioRender site to create high resolution schematic figures. Follow this link and enter your details and the manuscript number to create and download figures. Upload these as the figure files for your revised submission. If you choose not to take up this offer we require figures to be of similar quality and resolution. If you are opting out of this service to authors, state this in the Comments section on the Detailed Information page of the submission form. The link provided should only be used for the purposes of this submission. Authors will be charged for figures created on this premium BioRender account if they are not related to this manuscript submission.
- Please ensure that any tables are in Word format and are, wherever possible, embedded in the article file itself.
- Papers must comply with the Statistics Policy: https://jp.msubmit.net/cgi-bin/main.plex?form_type=display_requirements#statistics.

In summary:

- If $n \leq 30$, all data points must be plotted in the figure in a way that reveals their range and distribution. A bar graph with data points overlaid, a box and whisker plot or a violin plot (preferably with data points included) are acceptable formats.
- If $n > 30$, then the entire raw dataset must be made available either as supporting information, or hosted on a not-for-profit repository e.g. FigShare, with access details provided in the manuscript.
- 'n' clearly defined (e.g. x cells from y slices in z animals) in the Methods. Authors should be mindful of pseudoreplication.
- All relevant 'n' values must be clearly stated in the main text, figures and tables, and the Statistical Summary Document (required upon revision).
- The most appropriate summary statistic (e.g. mean or median and standard deviation) must be used. Standard Error of the Mean (SEM) alone is not permitted.
- Exact p values must be stated. Authors must not use 'greater than' or 'less than'. Exact p values must be stated to three significant figures even when 'no statistical significance' is claimed.
- Statistics Summary Document completed appropriately upon revision.

EDITOR COMMENTS

Reviewing Editor:

The revised version has greatly improved the manuscript yet it still falls short of meeting the requirements for acceptance. Please see reviewers detailed comments below. Specifically, please see if the paper can be made more concise.

REFEREE COMMENTS

Referee #2:

The revision improves many of the points that I found confusing in the original manuscript, and rebuts some of the points I made.

I think my general point stand, the paper has limited impact because of the nature of the subject: The technical difficulty of this sort of work means that data is limited. However, there is an advantage to lumping together data from "units" which can involve different types of neurons (here Purkinje cells, mossy fibre terminals and cerebral cortical interneurons, here putative Golgi cells). Combining this sort of data makes the groupings look larger, but data that emerge is mixed - for example after a couple of pages of description of receptive field testing the conclusion (starting on page 462) is that "in our sample of Purkinje cells the findings were mixed: for the majority the overlapping receptive fields is consistent with the hypothesis that the modulation of simple spike activity by sensory inputs occurs as a consequence of excitation of local granule cells: whereas for the remaining Purkinje cells the results are consistent with excitation from granule cells located in non-locally. However, clearly caution is needed in interpreting a small sample size, and further studies will be required to test these findings more fully."

This does however it represents the type of approach which is rarely undertaken nowadays And asks a fundamental question. Again due to the nature of the subject the conclusions are relatively broad.

Referee #3:

The authors addressed some of my concerns. I still have several remaining concerns.

>The animals remained passive during receptive field mapping and did not show any behavioural responses i.e. they would sit quietly during testing.

This information is not in the Methods. Please add it because no behavioral responses to stimulus is important.

>The experiments were carried out with at least two experimenters (and in many cases 3) who would reach consensus about the receptive field.

Please also include this information ideally with identity of the experimenters for verifying the reproducibility of the receptive field mapping experiments.

>As stated in results (p23, line 557) these previous studies in cats have shown that the reaching task used is highly stereotypical, with little variation between trials in terms of movement trajectories and velocity with cats adopting a very similar kinematic reaching strategy.

I agree that the limb movements in the previous studies are stereotypical. However, the authors did not use the data from these previous studies, thus shouldn't they show this with their own data? The authors analyzed the duration, i. e., the velocity in detail but did not analyze movement trajectories which would affect the neural activity.

>We focussed our analysis to the paw lift and reaching to the tube because other events later in the task such as tube exit, paw down etc were considerably more variable in timing.

This was what I was concerning about. I'm wondering if PETHs shown and used for clustering include these variable events. The authors should consider whether the neural activity of these later periods does not affect the results. Would the author reach the same conclusion if the period of PETHs used for analyses were limited for paw lift and tube entry?

>We have now included the PCA and k-means in Figure 8 and have added.

In Figure 8B, C, D, F, correspondence between colors and the cluster number should be indicated.

>Thus, we wanted to ask what is the clustering implied by using the same number of clusters for the spike trains as we believe are present because of the grouping of the units by receptive field.

This would still lead to the wrong conclusion. There are several indices for the number of clusters (elbow methods, silhouette coefficient, BIC score), so they should evaluate the results of clustering using one of these measures. Also, there is still no data regarding partially classified units. Table 1 still shows only the results of classification without any data.

END OF COMMENTS

1st Confidential Review

27-Jun-2022

We have addressed the comment's raised by Referee #3 below.

>The animals remained passive during receptive field mapping and did not show any behavioural responses i.e. they would sit quietly during testing.

This information is not in the Methods. Please add it because no behavioral responses to stimulus is important.

This has now been included in the Methods (Page 9, Line 232).

>The experiments were carried out with at least two experimenters (and in many cases 3) who would reach consensus about the receptive field.

Please also include this information ideally with identity of the experimenters for verifying the reproducibility of the receptive field mapping experiments.

We have now included this information in the Methods (page 9, Line 240).

>As stated in results (p23, line 557) these previous studies in cats have shown that the reaching task used is highly stereotypical, with little variation between trials in terms of movement trajectories and velocity with cats adopting a very similar kinematic reaching strategy.

I agree that the limb movements in the previous studies are stereotypical. However, the authors did not use the data from these previous studies, thus shouldn't they show this with their own data? The authors analyzed the duration, i. e., the velocity in detail but did not analyze movement trajectories which would affect the neural activity.

We have video recordings from one animal and have analysed the trajectories. Two days demonstrating the trajectories have been shown – at the start and the end of the recording life of the animal. Some caution needs to be made when comparing the trajectories as the frame rate of the video was 60 fps which makes tracking the paw difficult but also because the marker point on the paw varies between sessions. However, even with this in mind, in general, the trajectories are consistent within a session and between sessions (see Fig Supporting Information 1) and is similar to what Alstermark and colleagues have found. We have included this figure as Supporting Information and have added this to the results (page 23, Line 565).

>We focussed our analysis to the paw lift and reaching to the tube because other events later in the task such as tube exit, paw down etc were considerably more variable in timing.

This was what I was concerning about. I'm wondering if PETHs shown and used for clustering include these variable events. The authors should consider whether the neural activity of these later periods does not affect the results. Would the author reach the same conclusion if the period of PETHs used for analyses were limited for paw lift and tube entry?

The clustering uses the standardised z-score in relation to paw lift (time point 0 s) with an overall time duration of 2 s. We also carried out clustering using a shorter overall time duration and found no change in the clustering. As most of the neural activity occurred around the time of paw lift, this is not a surprising finding.

It therefore seems reasonable to conclude that the clustering relies primarily on neural activity during the period around paw lift. Because of the low number of cells responding to later events in the movement (e.g tube entry, paw down) we were unable to examine clustering in relation to these later periods. These points are now made in the Methods (Page 10, Line 273 & 291) and Results (Page 26, 617)

>We have now included the PCA and k-means in Figure 8 and have added.

We have included the legend for these panels and added details in the figure legend. .

>Thus, we wanted to ask what is the clustering implied by using the same number of clusters for the spike trains as we believe are present because of the grouping of the units by receptive field.

This would still lead to the wrong conclusion. There are several indices for the number of clusters (elbow methods, silhouette coefficient, BIC score), so they should evaluate the results of clustering using one of these measures. Also, there is still no data regarding partially classified units. Table 1 still shows only the results of classification without any data.

The data were clustered using k-means with the number of clusters chosen to match the number of available receptive field classes. Since our aim was to compare how well clustering by electrophysiological activity matches the receptive field classes we did not consider other cluster numbers, but measures intended to discover the appropriate number of clusters do not appear to distinguish this, or any other, number, as the best number of clusters.

In relation to Table 1: a) given the editor's request to make the manuscript more concise; b) the main aim of the study was investigate the organization of positively identified microzones within the C3 zone and their neural activity during a reaching task in awake animals; and c) the low sample size for the partially classified unit data, we have removed this additional analysis from Results. The conclusions of the study remain unchanged.

Dear Dr Cerminara,

Re: JP-RP-2022-282928R2 "Neuronal activity patterns in microcircuits of the cerebellar cortical C3 zone during reaching" by Nadia L Cerminara, Martin Garwicz, Henry Darch, Conor Houghton, Dilwyn E Marple-Horvat, and Richard Apps

I am pleased to tell you that your paper has been accepted for publication in The Journal of Physiology.

NEW POLICY: In order to improve the transparency of its peer review process The Journal of Physiology publishes online as supporting information the peer review history of all articles accepted for publication. Readers will have access to decision letters, including all Editors' comments and referee reports, for each version of the manuscript and any author responses to peer review comments. Referees can decide whether or not they wish to be named on the peer review history document.

The last Word version of the paper submitted will be used by the Production Editors to prepare your proof. When this is ready you will receive an email containing a link to Wiley's Online Proofing System. The proof should be checked and corrected as quickly as possible.

Authors should note that it is too late at this point to offer corrections prior to proofing. The accepted version will be published online, ahead of the copy edited and typeset version being made available. Major corrections at proof stage, such as changes to figures, will be referred to the Reviewing Editor for approval before they can be incorporated. Only minor changes, such as to style and consistency, should be made a proof stage. Changes that need to be made after proof stage will usually require a formal correction notice.

All queries at proof stage should be sent to TJP@wiley.com.

Are you on Twitter? Once your paper is online, why not share your achievement with your followers. Please tag The Journal (@jphysiol) in any tweets and we will share your accepted paper with our 23,000+ followers!

Yours sincerely,

Richard Carson
Senior Editor
The Journal of Physiology

P.S. - You can help your research get the attention it deserves! Check out Wiley's free Promotion Guide for best-practice recommendations for promoting your work at www.wileyauthors.com/eeo/guide. And learn more about Wiley Editing Services which offers professional video, design, and writing services to create shareable video abstracts, infographics, conference posters, lay summaries, and research news stories for your research at www.wileyauthors.com/eeo/promotion.

*** IMPORTANT NOTICE ABOUT OPEN ACCESS ***

To assist authors whose funding agencies mandate public access to published research findings sooner than 12 months after publication The Journal of Physiology allows authors to pay an open access (OA) fee to have their papers made freely available immediately on publication.

You will receive an email from Wiley with details on how to register or log-in to Wiley Authors Services where you will be able to place an OnlineOpen order.

You can check if your funder or institution has a Wiley Open Access Account here <https://authorservices.wiley.com/author-resources/Journal-Authors/licensing-and-open-access/open-access/author-compliance-tool.html>

Your article will be made Open Access upon publication, or as soon as payment is received.

If you wish to put your paper on an OA website such as PMC or UKPMC or your institutional repository within 12 months of publication you must pay the open access fee, which covers the cost of publication.

OnlineOpen articles are deposited in PubMed Central (PMC) and PMC mirror sites. Authors of OnlineOpen articles are permitted to post the final, published PDF of their article on a website, institutional repository, or other free public server, immediately on publication.

Note to NIH-funded authors: The Journal of Physiology is published on PMC 12 months after publication, NIH-funded authors DO NOT NEED to pay to publish and DO NOT NEED to post their accepted papers on PMC.

EDITOR COMMENTS

Congratulations! The revised version satisfactorily addresses all outstanding comments.

REFEREE COMMENTS

I have no more comments on the manuscript.

2nd Confidential Review

01-Sep-2022